# Feature Learning beyond the Lazy-Rich Dichotomy: Insights from Representational Geometry

Chi-Ning Chou [* 1]   Hang Le [* 1]   Yichen Wang [2]   SueYeon Chung [1 3]

## Abstract

Integrating task-relevant information into neural representations is a fundamental ability of both biological and artificial intelligence systems. Recent theories have categorized learning into two regimes: the rich regime, where neural networks actively learn task-relevant features, and the lazy regime, where networks behave like random feature models. Yet this simple lazy–rich dichotomy overlooks a diverse underlying taxonomy of feature learning, shaped by differences in learning algorithms, network architectures, and data properties. To address this gap, we introduce an analysis framework to study feature learning via the geometry of neural representations. Rather than inspecting individual learned features, we characterize how task-relevant representational manifolds evolve throughout the learning process. We show, in both theoretical and empirical settings, that as networks learn features, task-relevant manifolds untangle, with changes in manifold geometry revealing distinct learning stages and strategies beyond the lazy–rich dichotomy. This framework provides novel insights into feature learning across neuroscience and machine learning, shedding light on structural inductive biases in neural circuits and the mechanisms underlying out-of-distribution generalization.

## 1. Introduction

Learning induces changes in brain activity, whether it involves navigating a new city, adapting novel motor skills, or

[*]Equal contribution  [1]Center for Computational Neuroscience, Flatiron Institute, New York, NY, USA [2]University of California, UCLA, Los Angeles, CA, USA [3]Center for Neural Science, New York University, New York, NY, USA. Correspondence to: Chi-Ning Chou <cchou@flatironinstitute.org>, Hang Le <hle@flatironinstitute.org>, SueYeon Chung <schung@flatironinstitute.org>.

*Proceedings of the 42nd International Conference on Machine Learning*, Vancouver, Canada. PMLR 267, 2025. Copyright 2025 by the author(s).

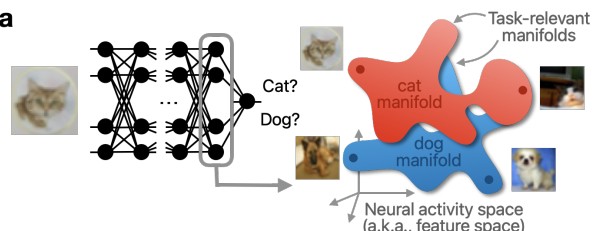

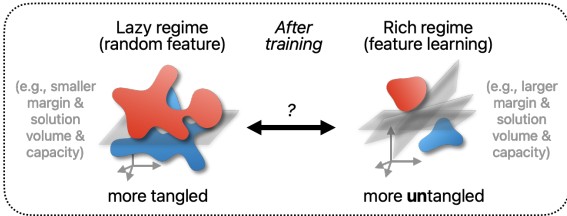

*Figure 1.* Schematic illustration. **a**, We propose to investigate feature learning using representational geometry and task-relevant manifolds. **b**, Feature learning (i.e., rich regime) can be viewed as a process of *untangling* task-relevant manifolds—structuring the neural activity space to improve separation among manifolds.

acquiring new cognitive tasks. These changes are reflected in the incorporation of task-relevant features into neural representations (Olshausen & Field, 1996; Poort et al., 2015; Niv, 2019; Reinert et al., 2021; Gurnani & Gajic, 2023). Similarly, the remarkable success of deep learning is often attributed to the ability of neural networks to learn problem-specific features[1]. For example, in deep neural networks (DNNs) (LeCun et al., 1998; Krizhevsky et al., 2012), the ability to learn rich feature hierarchies enables superior image classification performance (Girshick et al., 2014). Meanwhile, the seminal work of (Chizat et al., 2019) demonstrated that neural networks can perform well even when there are negligible changes in the weights of the networks. These observations raise important questions: Do neural networks always need to learn task-relevant features? How can we evaluate the quality of the features they learn?

[1]In this paper, *features* broadly refer to measurable properties or characteristics of patterns in data/input.

To answer these questions, researchers in representation learning have developed several methods to determine whether a neural network operates in the *lazy* regime (learning without changing internal features) or the *rich* regime (learning task-relevant features)[2]. These methods include measuring changes in the weights of the network, tracking activated neurons, and assessing differences in the linearized model (also known as the neural tangent kernel, NTK (Jacot et al., 2018)). Factors such as initial weight norm, learning rate, and readout weight have been found to play a role in whether a network is lazy or rich (Chizat et al., 2019). Moreover, recent theoretical evidence has suggested that networks could perform better in the rich regime compared to the lazy regime (Yang & Hu, 2021; Shi et al., 2022; Karp et al., 2021; Damian et al., 2022; Ba et al., 2022).

However, feature learning is much *richer* than the lazy versus rich dichotomy. For example, changes in representations are not always beneficial as they can lead to issues such as catastrophic forgetting (Kirkpatrick et al., 2017). Moreover, different network architectures, training procedures, objective functions, and initializations can result in different inductive biases for feature learning (Chizat et al., 2019; Bordelon & Pehlevan, 2022; Ba et al., 2022; Damian et al., 2022), yet all of these scenarios could fall under the broad category of rich learning. Lastly, limitations in neuroscience technology for tracking synaptic weight changes in neural circuits necessitate a framework based on neural activities rather than network weights or neural tangent kernel.

## 1.1. Contributions

We go beyond the lazy-versus-rich dichotomy and address the above-mentioned gaps by investigating feature learning though the geometric properties of task-relevant manifolds. Here, task-relevant manifolds refer to the point clouds of neural activity patterns that are related to the tasks. In classification, for instance, a manifold could be the point cloud of neural activations corresponding to a stimulus category (e.g., the cat and dog manifolds in Figure 1a, left). In other domains, a manifold could correspond to a context (e.g., environmental cues) in a neuroscience experiment or to a concept (e.g., semantic categories) in a language model.

Instead of explicitly identifying which features a network learns, we ask: what changes occur in task-relevant manifolds as a network undergoes feature learning?

> When a neural network learns useful features from data for solving some task(s), task-relevant manifolds become better organized and more separable.

From this perspective, feature learning can be viewed as a process of *untangling* tangled manifolds at initialization (i.e., random features) to facilitate easy separation.

To make this intuition concrete and quantitative, we use *manifold capacity*[3] (Chung et al., 2018; Chou et al., 2025) (Definition 2.1 and Definition B.3), to quantify the degree of richness in feature learning (Figure 2b). Additionally, we use geometric measures that are analytically connected to manifold capacity as mechanistic descriptors to explain how task-relevant manifolds untangle. To demonstrate our proposed method, we systematically study a wide range of settings, from theoretical models to machine learning and neuroscience problems. Our contributions include:

- (Section 3) We use manifold capacity as a representation-based method to quantify the degree of feature learning and demonstrate that it is better than conventional measures across a wide range of settings.

- (Section 4) Manifold geometry reveals previously unreported subtypes of feature learning. We find that the training of neural networks undergoes various *learning stages* as shown by the dynamics of manifold geometry, and there are emergent *learning strategies* as networks having different degree of richness in learning.

- (Section 5) We find new geometric insights that have not been reported in problems from neuroscience (e.g., structural inductive biases in neural circuits) and machine learning (e.g., out-of-distribution generalization).

## 1.2. Related work

Feature learning has been a fundamental research problem in various domains, including neuroscience and machine learning. In neuroscience, understanding the relationship between neural representations and task performance is a central focus (Gao & Ganguli, 2015). Representational geometry (Chung & Abbott, 2021) has emerged as a promising approach to investigate how different organizations of features can lead to better task performance (Bernardi et al., 2020; Flesch et al., 2022; Gurnani & Gajic, 2023). Several studies have explored methods for inferring the underlying learning rules of a neural network using representational geometry (Cao et al., 2020; Sorscher et al., 2022) and low-order statistics (Nayebi et al., 2020). In machine learning, *visualization techniques* (Zeiler & Fergus, 2014) have been widely used to gain intuitive insights into learned represen-

---

[2]These two regimes are also known as *kernel regime* and *feature learning regime* .

[3]Manifold untangling originates from neuroscience (DiCarlo & Cox, 2007) and refers to the intuition that task-relevant manifolds become increasingly separable in a high-dimensional state space. While there have been methods (Yamins & DiCarlo, 2016; Hong et al., 2016) using *worst-case* decoding accuracy to quantify the degree of manifold untangling, manifold capacity is an *average-case* method that is known to be better in capturing complex shapes of manifolds. See Section 2.2 and Appendix B for more details.

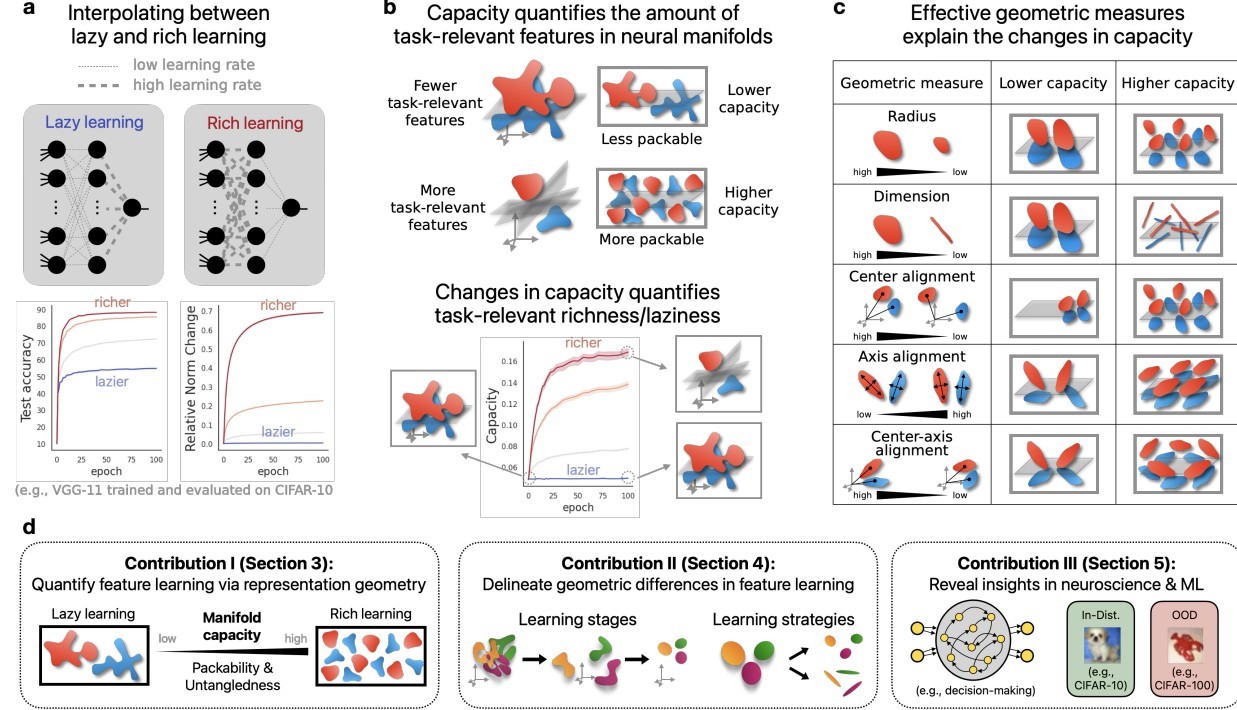

*Figure 2.* Our methods. **a**, We adopt the method from (Chizat et al., 2019) which interpolates lazy (in blue) and rich (in red) learning via adjusting a scale factor of learning rate (see Section 2.1). **b**, We propose to use changes of capacity across training to study task-relevant richness/laziness in feature learning (Section 3). **c**, Effective geometric measures are analytically connected to the capacity value (Chou et al., 2025), providing mechanistic descriptors to study representational changes in feature learning. **d**, Our three main contributions.

tations. On the theoretical front, the *kernel method* (Jacot et al., 2018; Lee et al., 2019) has been a leading approach to analytically characterize the behavior of neural networks, particularly in terms of their deviation from the corresponding kernel. This line of research includes studies on the distinction between lazy and rich regimes (Chizat et al., 2019; Geiger et al., 2020) and identifying problem settings where neural networks with feature learning outperform kernel methods (Ba et al., 2022; Dandi et al., 2023; Yang & Hu, 2021). See Appendix A for more on related work.

## 2. Method and Setup

### 2.1. Rich and lazy training in neural networks

We studied rich versus lazy learning in two standard settings: 2-layer non-linear neural networks on synthetic data and feedforward deep neural networks on real image classification datasets (Chizat et al., 2019). All analyses were performed on the test data representations in the last layer.

**Interpolating between rich and lazy regime.** In all experiments, we use the *inverse scale factor* $\bar{\eta}$ as a tunable ground truth for the degree of feature learning. In particular, $\bar{\eta}$ controls the magnitude of the output of the network, as in (Chizat et al., 2019). Intuitively, a larger $\bar{\eta}$ leads to

higher learning rate of intermediate layers compared to that of the readout weights, resulting in a richer learning process (Figure 2a). See Appendix D and Appendix E for details.

**Neural networks.** For 2-layer networks, we considered synthetic data models with random point clouds as input manifolds. This setting serves as a testbed for testing the proposed method and building intuitions. See Appendix D for details. For feedforward DNNs, as the goal of this work is to understand neural representations rather than pushing benchmarks, we focused on models and settings that are large enough to see interesting phenomena. Specifically, we considered VGG-11 (Simonyan & Zisserman, 2015) and ResNet-18 (He et al., 2016) and datasets CIFAR-10 (Krizhevsky & Hinton, 2009), CIFAR-100 (Krizhevsky & Hinton, 2009), CIFAR-10C (Hendrycks & Dietterich, 2018). See Appendix E for details. In all reported experimental results, we trained 5 models initialized with different seed.

**Task-relevant manifolds.** Let $P$ be the number of classes and $N$ be the number of neurons (in the layer of interest, e.g., the last layer). The $i$-th class manifold is modeled as the convex set (as in linear classification, it is mathematically equivalent to study the convex hull of a manifold) $\mathcal{M}_i = \mathsf{conv}(\{\Phi(x) : x \in \mathcal{X}_i\})$ where $\mathcal{X}_i$ is the collection of inputs in the $i$-th class, $\Phi(x)$ is the representation for $x$, and $\mathsf{conv}(\cdot)$ denotes the convex hull.

## 2.2. Manifold capacity theory

Manifold capacity theory (Chung et al., 2018; Chou et al., 2025) extends the classic notion of storage capacity of points (Cover, 1965; Gardner & Derrida, 1988; Gardner, 1988) to object manifolds for study the *untangling hypothesis*[4] of invariant object recognition in vision neuroscience (DiCarlo & Cox, 2007). Manifold capacity quantifies manifold packability in terms of linear classification, making it a useful metric for assessing the degree of untangling.

**Definition 2.1** (Simulated manifold capacity (Chou et al., 2025)). Let $P, N \in \mathbb{N}$ and $\mathcal{M}_i \subseteq \mathbb{R}^N$ be convex sets for each $i \in [P] = \{1, \ldots, P\}$. For each $n \in [N]$, define

$$p_n := \Pr_{\mathbf{y}, \Pi_n} [\exists \theta \in \mathbb{R}^n : y_i \langle \theta, \mathbf{s} \rangle \geq 0, \forall i \in [P], \mathbf{s} \in \mathcal{M}_i)]$$

where $\mathbf{y}$ is a dichotomy vector randomly sampled[6] from $\mathcal{Y} \subset \{\pm 1\}^P$ and $\Pi_n$ is a random projection from $\mathbb{R}^N$ to $\mathbb{R}^n$. Suppose $p_N = 1$ (which means that the input manifolds are linearly separable in $\mathbb{R}^N$), the simulated capacity of $\{\mathcal{M}_i\}_{i \in [P]}$ is defined as

$$\alpha_{\text{sim}} := \frac{P}{\sum_n (1 - p_n)}.$$

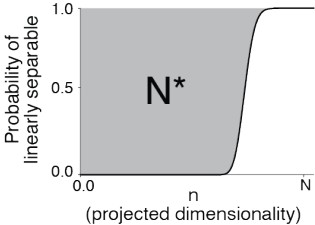

*Figure 3.* The area above the curve of linearly separable probability is the critical dimension $N^* := \sum_n (1 - p_n) \approx \min_{n : p_n \geq 0.5} \{n\}$, which appears in the denominator of the definition for $\alpha_{\text{sim}}$.

Intuitively, the simulated manifold capacity measures the *packability* (Chung et al., 2018) of manifolds by determining the smallest dimensional subspace needed to ensure that the manifolds can be separated. Namely, manifolds that are more packable (i.e., separable when projected to smaller dimensional subspaces) exhibit higher manifold capacity. While Definition 2.1 can be numerically simulated (Algorithm 1), a recent theory, *Geometry Linked to Untangling Efficiency (GLUE)* (Chou et al., 2025) further derived an analytical formula through techniques from statistical physics

---

[4]The "untangling hypothesis" posits that the brain transforms complex, entangled sensory inputs into more linearly separable representations, facilitating efficient object recognition.

[6]Typically we pick $\mathcal{Y}$ to be the collection of all 1-versus-rest dichotomies.

[6]See Figure 4 for how NTK-label alignment and representation-label alignment could fail at quantifying task-relevant features.

and convex geometry (Definition B.3):

$$\alpha_M^{-1} = \frac{1}{P} \mathbb{E}_{\substack{\mathbf{y} \sim \mathcal{Y} \\ \mathbf{t} \sim \mathcal{N}(0, I_N)}} \left[ \max_{\mathbf{s}_i \in \mathcal{M}_i} \left\{ \|\text{proj}_{\text{cone}(\{y_i \mathbf{s}_i\})} \mathbf{t}\|_2^2 \right\} \right] \quad (1)$$

where $\mathcal{N}(\cdot, \cdot)$ denotes the multivariate Gaussian distribution and $\text{cone}(\cdot)$ is the convex cone spanned by the vectors, i.e., $\text{cone}(\{y_i \mathbf{s}_i\}) = \{\sum_i \lambda_i y_i \mathbf{s}_i : \lambda_i \geq 0\}$. Specifically, $|\alpha_{\text{sim}} - \alpha_M| = O(1/N)$ and $\alpha_M$ can be computed by solving quadratic programs (Algorithm 2).

## 2.3. Effective geometric measures

Equation 1 connects manifold capacity to the structure of the manifolds $\{\mathcal{M}_i\}$. Concretely, for each $\mathbf{y}, \mathbf{t}$, define $\{\mathbf{s}_i(\mathbf{y}, \mathbf{t})\} = y_i \cdot \arg\max_{\{\mathbf{s}_i\}} \|\text{proj}_{\text{cone}(\{y_i \mathbf{s}_i\})} \mathbf{t}\|_2^2$ as the *anchor points* with respect to $\mathbf{y}$ and $\mathbf{t}$. Intuitively, these anchor points are the support vectors with respect to some random projection $\mathbf{t}$ and dichotomy $\mathbf{y}$. Namely, the randomness induces a distribution of anchor points supported on the manifolds $\{\mathcal{M}_i\}$; manifold capacity then emerges naturally as a summary statistic derived from this distribution. This connection between manifold capacity and anchor points motivated previous work (Chung et al., 2018; Chou et al., 2025) to define the following effective manifold geometric measures that capture the structure of manifolds while being analytically connected to capacity (see Figure 2c and Appendix B).

**Definition 2.2** (Effective manifold geometric measures (Chung et al., 2018; Chou et al., 2025), informal). For each $i \in [P]$, define $\mathbf{s}_i^0 := \mathbb{E}_{\mathbf{y}, \mathbf{t}}[\mathbf{s}_i(\mathbf{y}, \mathbf{t})]$ as the **center** of the $i$-th manifold and define $\mathbf{s}_i^1(\mathbf{y}, \mathbf{t}) := \mathbf{s}_i(\mathbf{y}, \mathbf{t}) - \mathbf{s}_i^0$ to be the **axis** part of $\mathbf{s}_i(\mathbf{y}, \mathbf{t})$ for each pair of $(\mathbf{y}, \mathbf{t})$.

- **Manifold dimension** captures the degree of freedom of the noises/variations within the manifolds. It is approximately $D_M \approx \mathbb{E}_{\mathbf{y}, \mathbf{t}} \left[ \frac{1}{P} \sum_i \left( \frac{\langle \mathbf{s}_i^1(\mathbf{y}, \mathbf{t}), \mathbf{t} \rangle}{\|\mathbf{s}_i^1(\mathbf{y}, \mathbf{t})\|_2} \right)^2 \right]$, which is analogous to the Gaussian width of the manifolds (Vershynin, 2018, Section 7.7).

- **Manifold radius** captures the noise-to-signal ratio of the manifolds. It is approximately $R_M \approx \mathbb{E}_{\mathbf{y}, \mathbf{t}} \left[ \frac{1}{P} \sum_i \frac{\|\mathbf{s}_i^1(\mathbf{y}, \mathbf{t})\|_2^2}{\|\mathbf{s}_i^0\|_2^2} \right]$.

- **Center alignment** captures the correlation between the center of manifolds and is defined as $\rho_M^c := \frac{1}{P(P-1)} \sum_{i \neq j} |\langle \mathbf{s}_i^0, \mathbf{s}_j^0 \rangle|$.

- **Axis alignment** captures the correlation between the axis of manifolds and is defined as $\rho_M^a := \frac{1}{P(P-1)} \sum_{i \neq j} \mathbb{E}_{\mathbf{y}, \mathbf{t}}[|\langle \mathbf{s}_i^1(\mathbf{y}, \mathbf{t}), \mathbf{s}_j^1(\mathbf{y}, \mathbf{t}) \rangle|]$.

- **Center-axis alignment** captures the correlation between the center and axis of manifolds and is defined as $\psi_M := \frac{1}{P(P-1)} \sum_{i \neq j} \mathbb{E}_{\mathbf{y}, \mathbf{t}}[|\langle \mathbf{s}_i^0, \mathbf{s}_j^1(\mathbf{y}, \mathbf{t}) \rangle|]$.

*Table 1.* Comparison to conventional measures used in lazy versus rich learning.

| | Our approach (manifold geometry) | Accuracy | Weight changes | NTK-label alignment | Representation-label alignment |
|---|---|---|---|---|---|
| Detect the changes in features | ✓ | × | ✓ | ✓ | ✓ |
| Quantify the amount of task-relevant features | ✓ | × | × | ×[5] | ×[5] |
| Representation-based | ✓ | × | × | × | ✓ |
| Delineate subtypes of feature learning | ✓ | × | × | × | × |

Three important remarks on effective manifold geometric measures to be made: First, the changes in manifold capacity can be explained by the changes of these geometric measures. For example, the decrease of manifold radius and dimension makes the capacity higher (see Figure 2c, Section B.4). Second, these effective geometric measures faithfully track the corresponding underlying geometric properties in well-studied mathematical settings (see Section B.5). Moreover, there is a simple formula connecting manifold capacity with effective geometric measure: $\alpha_M \approx (1 + R_M^{-2})/D_M$ (see Appendix B for details). Finally, combining the above two points, these effective geometric measures serve as intermediate-level descriptors to investigate how different structural properties of neural manifolds contribute to the changes of task-level performance.

## 3. Manifold capacity quantifies the degree of feature learning

In this section, we provide both empirical and theoretical justifications for using the increase in capacity during training as a measure to quantify the degree of richness (or the amount of task-relevant features) in feature learning. Furthermore, we compare our method with conventional approaches in the study of lazy versus rich learning, highlighting the new insights uncovered by our approach.

### 3.1. Justifications of capacity for quantifying the lazy versus rich dichotomy

**Empirical justification in standard settings.** We start with empirically justifying the use of capacity to quantify the degree of feature learning. A classic result in the literature of lazy versus rich training is to train a lazy network where the test accuracy improves, but the weight matrices (or kernels) do not change much before and after training. We consider two settings in (Chizat et al., 2019), one is feedforward DNNs (VGG-11 and ResNet-18) trained on CIFAR-10 (Figure 2a-b, Figure 15, and Figure 14), and the other is 2-layer non-linear NNs trained on random point clouds (Figure 4a). In both cases, we observe that the manifolds are more untangled when training is richer and capacity correctly tracks the degree of feature learning (the ground truth is $\bar{\eta}$). This provides empirical justification for the use of capacity and manifold untangling to quantify feature learning.

**Theoretical justification on 2-layer non-linear networks.** To strengthen the connection between capacity and feature learning, we next consider a well-studied theoretical model (Ba et al., 2022; Montanari et al., 2025) and analytically characterize the relationship between capacity, prediction error, and the effective degree of richness. Concretely, we consider the training of a fully-connected 2-layer network of the form $f(\mathbf{x}) = \frac{1}{\sqrt{N}}\mathbf{a}^\top \sigma(W^\top \mathbf{x})$, where $\mathbf{x} \in \mathbb{R}^d$ is an input, $W \in \mathbb{R}^{N \times d}$ is the hidden layer matrix, $\mathbf{a} \in \mathbb{R}^N$ is the readout weight, and $\sigma : \mathbb{R} \to \mathbb{R}$ is the (non-linear) activation function. To study feature learning in this setting, it is common to consider $W$ to be randomly initialized (i.e., random feature model (Rahimi & Recht, 2007)) and update via gradient descent with squared loss. Meanwhile, the readout weight $\mathbf{a}$ is randomly initialized and fixed to avoid lazy learning (where the network minimally adjusts the hidden layer and focuses on learning a good readout weight) as well as enable mathematical analysis (Ba et al., 2022). Input data and label $(\mathbf{x}_1, y_1), \ldots, (\mathbf{x}_{P_{\text{train}}}, y_{P_{\text{train}}})$ were randomly generated by a teacher-student setting, where there is a hidden signal direction $\beta^*$ that correlates with the label (see Setting C.2 for the full setting). As previously proved in (Ba et al., 2022) (see Theorem C.5), in the proportional asymptotic limit (i.e., $P_{\text{train}}, d, N \to \infty$ at the same rate), the first-step gradient update can be approximated by a rank-1 matrix that contains label information, resulting in the updated weight to be more aligned with the hidden signal $\beta^*$. Hence, in this setting, the learning rate $\eta$ can be used as the ground-truth to measure the amount of task-relevant information (i.e., richness in learning) in the model representation after gradient updates.

We extend the previous results in (Ba et al., 2022) from a regression setting to a classification setting. Specifically, We prove that capacity correctly tracks the effective degree of richness after one gradient step[7]. Moreover, we derive a monotone connection between capacity and prediction accu-

---

[7]Here we follow the convention in (Ba et al., 2022) and study only the first gradient step as the key Gaussian equivalence step might not hold for more steps as remarked in footnote 2 of (Ba et al., 2022).

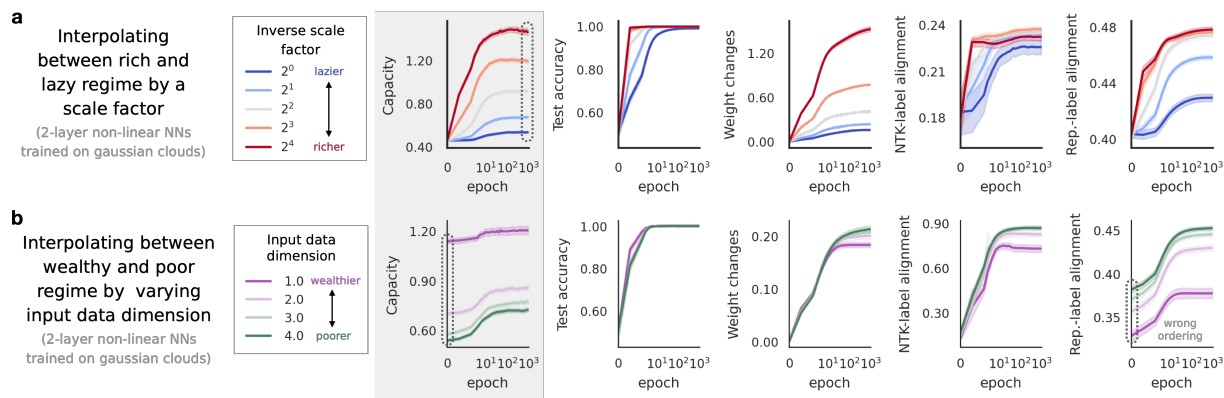

*Figure 4.* Capacity as a measure for the degree of feature learning. See Section D.1 for details. **a**, We interpolated between lazy and rich regime in 2-layer NNs trained to classify Gaussian clouds. We found that capacity could tell the difference between the underlying scale parameter better than the other conventional methods. **b**, We fixed a scale parameter and initialized the input Gaussian clouds with different dimensions (higher dimension leads to poorer initial representations). We found that capacity could tell the difference in the amount of tasks-relevant features at initialization. Other conventional methods, e.g., the representation-label alignment, could characterize the wrong ordering of wealthiness in initial features.

racy. Here, we provide an informal statement of our results and leave the formal version and proof in Appendix C.

**Theorem 3.1.** *Given Theorem C.1 and Setting C.2. Let $0 < \eta < \infty$ be the learning rate of a one-step gradient descent with squared loss and $\psi_1 = \frac{N}{d}, \psi_2 = \frac{P_{train}}{d}$ where $P_{train}$ is the number of training points, $d$ is the input dimension, and $N$ is the number of hidden neurons. Let $\alpha_{P_{train},d,N}(\eta)$ be the capacity and let $Acc_{P_{train},d,N}(\eta)$ be the prediction accuracy after a gradient step with learning rate $\eta$. We have*

1. *(Capacity tracks the degree of richness)*
   $\alpha_{P_{train},d,N}(\eta) \xrightarrow{P_{train},d,N\to\infty} \alpha(\eta,\psi_1,\psi_2)$ *where $\alpha(\cdot,\cdot,\cdot)$ is defined in Theorem C.4. Specifically, $\alpha(\eta,\psi_1,\psi_2) < \alpha(\eta',\psi_1,\psi_2)$ for every $0 < \eta < \eta'$.*

2. *(Capacity links to prediction accuracy)*
   $Acc_{P_{train},d,N}(\eta) \xrightarrow{P_{train},d,N\to\infty} Acc(\eta,\psi_1,\psi_2)$ *where $Acc(\eta,\psi_1,\psi_2)$ is formally defined in Theorem C.4. In particular, there exists an increasing and invertible function $h_{\psi_1,\psi_2} : \mathbb{R}_+ \to [0,1]$ such that $Acc(\eta,\psi_1,\psi_2) = h_{\psi_1,\psi_2}(\alpha(\eta,\psi_1,\psi_2))$.*

The above theorem justifies the usage of capacity as a measure for the degree of richness in feature learning within a well-studied theoretical setting. We remark that our proof requires substantial technical improvements from (Ba et al., 2022) due to the difference between regression and classification (e.g., analyzing the margin of the Gaussian equivalent model after one-step gradient using tools from (Montanari et al., 2025), Theorem C.7).

### 3.2. Comparison to conventional methods

Here we compare our method with several common measures for feature learning: accuracy curves, weight changes, and alignment methods (Table 1). Concretely, weight changes at the $t$-th epoch is defined as $\|W_t - W_0\|_F / \|W_0\|_F$ where $W_t$ is the weight matrix at the $t$-th epoch. NTK-label alignment and representation-label alignment at the $t$-th epoch are defined as $\mathrm{CKA}(K_t^{\mathrm{NTK}}, \mathbf{yy}^\top)$ and $\mathrm{CKA}(X_t X_t^\top, \mathbf{yy}^\top)$ respectively, where $\mathbf{y}$ is the label vector, $\mathrm{CKA}(\cdot,\cdot)$ is the center kernel alignment measure (Kornblith et al., 2019), $K_t^{\mathrm{NTK}}$ is the neural tangent kernel and $X_t$ is the representational matrix at the $t$-th epoch. To test these measures in a wide variety of settings, we consider 2-layer NNs with synthetic data where we can vary a wide range of parameters. See Appendix A for details about these methods and Appendix D for details about experiments.

**Capacity can detect task-relevant features in the presence of complex structures in data (Figure 4a).** We consider 2-layer NNs trained on random Gaussian clouds with gradient descent. We vary the scale parameter of the network to interpolate between lazy and rich regimes as done in (Chizat et al., 2019). We find that capacity is better at telling the difference of effective richness (i.e., the scale parameter) of the training than other conventional measures (Figure 4a). In particular, when the training is richer, we expect the representations to exhibit more complex structures. Manifold capacity excels at extracting task-relevant structures in representations because it is data-driven and free from additional statistical assumptions on the data.

**Capacity can quantify the differences in task-relevant features at initialization (Figure 4b).** When comparing two networks with different initializations, focusing solely on network changes can overlook differences in features present at initialization. Here, we use the capacity value at initialization to determine whether a network is in a wealthy

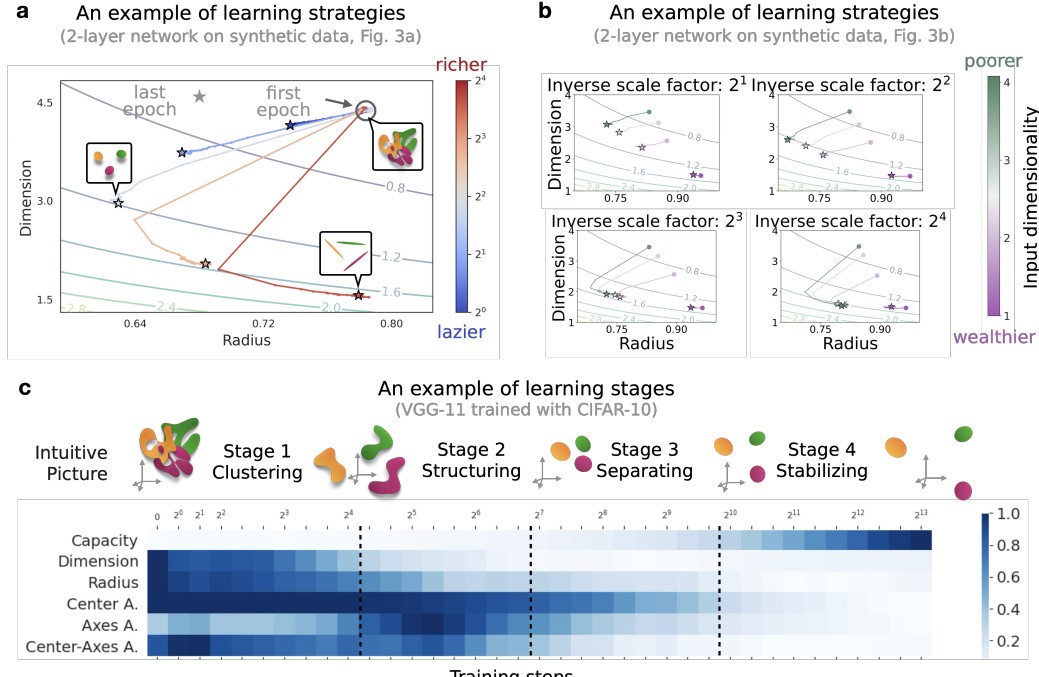

*Figure 5.* Manifold geometry characterizes learning strategies and learning stages. **a**, Capacity contour plot of the example from Figure 4a. The x-axis is the average manifold radius $R_M$, the y-axis is the average manifold dimension $D_M$, and the contour is the geometric approximation of capacity, i.e., $\alpha_M \approx (1 + R_M^{-2})/D_M$ (see Appendix B). **b**, Capacity contour plot of the example from Figure 4b. **c**, Normalized manifold geometry dynamics plot of VGG-11 trained with CIFAR-10. The values in each row are rescaled so that the min and max values are 0 and 1.

regime (i.e., possessing more task-relevant features) or a poor regime (i.e., possessing less task-relevant features), as shown in (Figure 4b). For example, a network is in a wealthy (resp. poor) regime when the manifolds are lower (resp. higher) dimensional because there are more (resp. less) features that are useful for telling the manifolds apart. The wealthy versus poor distinction provides insight into the network's initial state, allowing for a more comprehensive comparison of different settings (see Section 5.1).

## 4. Manifold Geometry Reveals Subtypes of Feature Learning

In this section, we demonstrate that feature learning is much richer than the lazy versus rich dichotomy. In particular, we use manifold geometric measures (Figure 2c, and Appendix B for details) to delineate the differences in the learned features (learning strategies) and representational changes throughout training (learning stages). The key takeaway from this section is the ability of our method to reveal task-relevant changes in neural representations.

### 4.1. Geometric differences in learned features: Learning strategies

To increase capacity, a network can shrink the radius and/or compress the dimension of neural manifolds (Figure 2c).

We demonstrate in 2-layer NNs the emergence of distinct learning strategies driven by different factors of training. In Figure 5a, we consider 2-layer networks trained on Gaussian clouds (as in Figure 4a). As training moves from the lazy to a richer regime (blue to gray), the network compresses both the radius and dimension to increase capacity. Interestingly, in an even richer regime (gray to red), the network sacrifices radius to further reduce dimension.

In Figure 5b, we consider the setting in Figure 4b where we interpolate the wealth of initialization by varying input data dimension. For the wealthiest initialization (purple), the network primarily compresses radius. For poorer initialization (green), both radius and dimension are compressed in lazier training, while in the richer regime (e.g., inverse scale factor $2^4$), the network sacrifices radius for further dimension compression. In summary, varying degrees of richness in feature learning can exhibit different learning mechanisms, as captured by manifold geometry.

### 4.2. Geometric changes during training: Learning stages

Neural networks learn in a highly non-monotonic manner throughout the training period. Examples include double descent (Belkin et al., 2019; Nakkiran et al., 2021; Mei & Montanari, 2022) and grokking (Power et al., 2022; Liu

et al., 2022; Nanda et al., 2023; Kumar et al., 2024). Previous works have analytically or empirically described the different stages/phases such as comprehension, grokking, memorization, and confusion (Liu et al., 2022) through the trajectory of accuracy curves.

From Figure 5a,b we observe distinct stages of manifold geometry evolution during training in 2-layer networks. In the very rich regime, the network initially compresses both radius and dimension, then increases radius to further reduce dimension. In Figure 5c, we examine a standard setting where VGG-11 is trained on CIFAR-10. Despite the rapid saturation of training and test accuracy, at least four stages of geometric changes are evident (see Figure 2c for analytical connections between geometric measures and capacity): a *clustering stage* (initial manifold compression), followed by a *structuring stage* (increasing alignment), a *separating stage* (decreasing alignment to push manifolds apart), and a final *stabilizing stage* (further reducing center alignment).

## 5. Applications to Neuroscience and Machine Learning Problems

In previous sections, we used capacity to quantify the degree of feature learning and delineate the learning stages and strategies through effective geometry. In this section, we apply our framework to find geometric insights in problems from neuroscience and machine learning.

### 5.1. Structural inductive biases in neural circuits

Structural variations (e.g., neural connectivity pattern) can play a crucial role in learning dynamics (Campagnola et al., 2022; Goudar et al., 2023; Xie et al., 2023; Raman & O'Leary, 2021). Theoretical studies have shown that certain patterns of neural connections can enhance the speed and/or performance at which specific tasks are learned (Canatar et al., 2021; Flesch et al., 2021; Braun et al., 2022). However, there is a gap between theory and experiments as currently there is no experimental technology that can easily measure the precise changes in synaptic weight of multiple neurons simultaneously. Meanwhile, studying the changes in neural activity can provide an alternative lens to probe the functional changes in neural circuits. We study recurrent neural networks (RNNs) that are trained on standard neuroscience tasks such as perceptual decision making (Britten et al., 1992) (Figure 6a). We adopt the setting from previous work (Liu et al., 2024) on investigating how differences in connectivity initialization affect the learning process.

**Experimental setup.** We use the `neurogym` package (Molano-Mazon et al., 2022) to simulate common cognitive tasks, e.g., perceptual decision making. To study how connectivity structure impacts learning strategies, we follow the setup in (Liu et al., 2024) and initialize recurrent neural net-

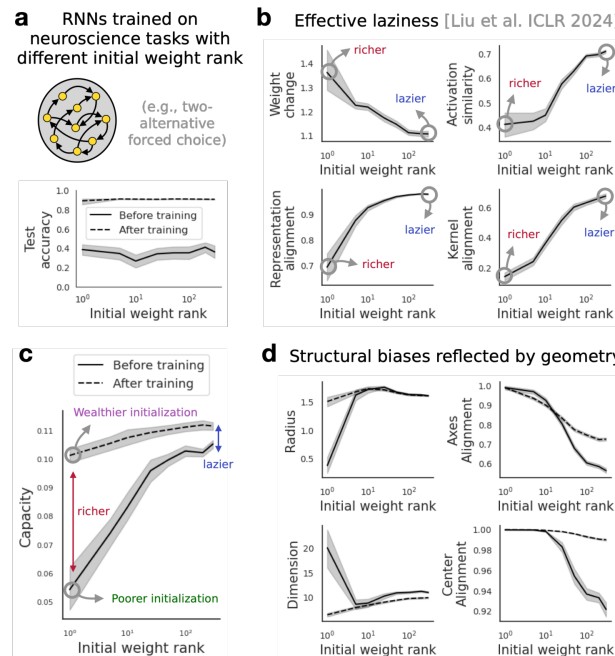

*Figure 6.* Structural inductive biases in neural circuits. **a**, We consider RNNs trained on standard neuroscience tasks. **b**, Previous work (Liu et al., 2024) found that the initial weight rank of the recurrent connectivity matrix leads to an inductive bias toward effectively richer or lazier training. **c**, We find that RNNs trained with different initial weight rank reach the same capacity value at final epoch. It is the difference in capacity at initialization that makes RNNs with small initial weight rank richer in training. **d**, Despite having the same capacity at final epoch, RNNs with different initial weight rank exhibit different manifold geometry.

works (RNN) weights with varying ranks via Singular Value Decomposition. The RNN have 300 hidden units, 1 layer, with ReLU activations, and are trained for 10000 iterations using `SGD` optimizer. Manifold capacity and effective geometric measures are computed using representations from the hidden states. See Appendix F for details.

**Our findings.** First, we study the training dynamics of capacity value in RNNs with various initial weight rank (Figure 6c). In agreement with the previous finding in (Liu et al., 2024) using weight changes, we find that the capacity changes of the small initial weight rank RNNs are higher than those of the large initial weight rank RNNs. Interestingly, the capacity values at the final epoch are about the same for RNNs with different initial weight rank. It is the difference in capacity value at initialization that distinguishes the learning dynamics of RNNs with different initial weight rank. Namely, small initial weight rank RNNs are in the poorer-richer feature learning regime, while large initial weight rank RNNs are in the wealthier-lazier feature learning (Figure 6c).

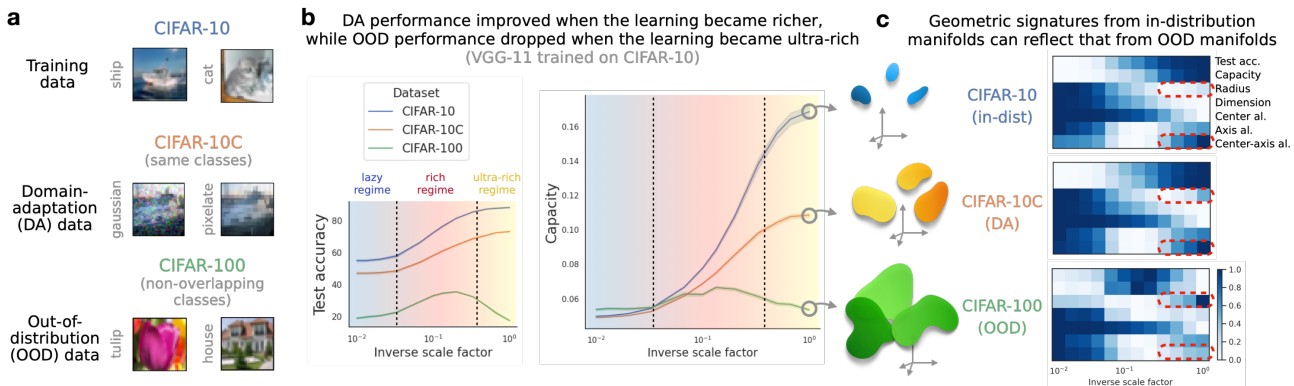

*Figure 7.* Out of distribution generalization. **a**, CIFAR-10C as a domain adaptation (DA) dataset and CIFAR-100 as an OOD dataset. **b**, Test accuracy improves for CIFAR-10 and CIFAR-10C as the training becomes richer and richer while the linear probe accuracy for CIFAR-100 would drastically drop in the ultra-rich training regime. **c**, Effective manifold geometry of CIFAR-100 reveals that the expansion of manifold radius and the increase of center-axis alignment explain the failure of OOD generalization in the ultra-rich regime. The color is normalized for each row respectively.

Next, we find that manifold geometric organizations are quite different for different initial weight ranks (Figure 6d). For example, poorer-richer learning (i.e., small initial weight rank) ends up with a larger radius and smaller dimension, while it is the opposite for wealthier-lazier learning (i.e., large initial weight rank). This finding suggests that RNNs exhibit structural biases at the manifold geometry level.

### 5.2. Out-of-distribution generalization

Understanding how neural networks learn and represent features is crucial not only for task performance but also for their ability to generalize beyond the training distribution. Out-of-distribution (OOD) generalization remains a fundamental challenge, as models often struggle when encountering data that deviates from their training set.

**Experimental setup.** For each model pre-trained on CIFAR-10, we train a linear classifier (i.e., linear probe (Alain & Bengio, 2016)) on top of the last-layer representation with CIFAR-100 train set, and then evaluate the linear probe's performance on CIFAR-100 test set (see details in Appendix Section E.4). We also consider a corrupted version of CIFAR-10, the CIFAR-10C dataset (Hendrycks & Dietterich, 2018) as an example of domain adaptation (DA) task. Finally, we compute the manifold capacity and effective geometric measures on last-layer representations.

**Our findings.** We see that the test accuracy of the OOD dataset increases when the network enters the rich learning regime ($\bar{\eta}$ around 0.1) but decreases drastically when the degree of feature learning is too rich ($\bar{\eta}$ around 1.0). The failure in such *ultra-rich* feature learning regime is different from the test accuracy of both CIFAR-10 and CIFAR-10C ( Figure 7b). Looking at the capacity and effective geometry ( Figure 7c), we first see strong correlations between the

capacity and test accuracy, which warrants the use of effective geometry. Next, we find that the expansion of manifold radius and the increase of center-axis alignment in the ultra-rich regime explain the drop of capacity. Interestingly, we also see an architectural difference where it is the increment in dimension in the ultra-rich regime explaining the drop of capacity in ResNet-18 (Figure 17). We leave it as a future direction to extend our study, applying these geometric insights to improve OOD generalization performance.

## 6. Conclusion and Discussion

Feature learning is a crucial *feature* in the study of neural networks in computational neuroscience and machine learning, and it is much *richer* than the lazy versus rich dichotomy. Understanding the connection between feature learning and performance paves the way for designing network architectures and learning algorithms with greater reliability and transparency for practical applications.

The primary contribution of this work is to demonstrate how the perspective of task-relevant manifold untangling (quantified by manifold capacity and delineated by manifold geometric measures) can enhance our understanding of feature learning at an intermediate level. We propose several promising future directions, including extending the theoretical analysis, exploring applications in other types of DNN (e.g., transformers) and addressing relevant scientific inquiries in neuroscience, such as inferring plasticity mechanisms from observed learning dynamics in neural data, and predicting learning-induced changes across brain regions. We believe that investigations in these intermediate-level understandings can be leveraged to design more robust, generalizable, and safer deep neural networks, as well as more accurate models for neuroscience applications.

## Acknowledgments

We thank William Yang, Yao-Yuan Yang, and the members of Chung lab for the discussion regarding many preliminary results and early versions of the manuscript. We thank Ghana Shyam Bandi, Shiyu Ling, Shreemayi Sonti, and Zoe Xiao for helpful discussion during the piloting stage of this project. This work was supported by the Center for Computational Neuroscience at the Flatiron Institute, Simons Foundation, and by NIH grant R01DA059220. S.C. was partially supported by a Sloan Research Fellowship, a Klingenstein-Simons Award, and the Samsung Advanced Institute of Technology project, "Next Generation Deep Learning: From Pattern Recognition to AI." All experiments were performed using the Flatiron Institute's high-performance computing cluster.

## Impact Statement

This paper presents work whose goal is to advance the field of Machine Learning. There are many potential societal consequences of our work, none which we feel must be specifically highlighted here.

## Code Availability

All code required to reproduce the figures presented is available under an MIT License at https://github.com/chung-neuroai-lab/feature-learning-geometry

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

# A. More on Related Work

**Visualization.**   Due to the black-box and complex nature of deep neural networks, various visualization techniques have been developed to attempt to characterize the features that models learn during training (*feature visualization*) and identify which input pixel and / or feature activation in the hidden layers contribute significantly to the final model outputs (*feature attribution*). *Feature visualization* techniques visualize features (e.g convolutional filter in the case of CNNs) by generating the input sample that maximizes the activation of that given feature via gradient descent (Olah et al., 2017) (Erhan et al., 2009) (Zeiler & Fergus, 2014). With its vivid visualization, *feature visualization* provide good intuition about the qualitative characteristics of the features that DNNs learn across layers (Zeiler & Fergus, 2014) as well as different types of models (e.g, standard vs adversarially robust  (Engstrom et al., 2019)). *Feature attribution* techniques generally identify how much each input and/or hidden features contribute to the final model prediction by computing the gradient of that input/hidden features to the output (some example techniques include saliency map (Simonyan et al., 2014), Grad-cam (Selvaraju et al., 2017), integrated gradient (Sundararajan et al., 2017)). Although both *feature visualization* and *feature attribution* offer intuitive understanding about the model's feature characteristics, the qualitative nature of visualization makes it difficult to quantify the degree of relevance of the learned features to a given task.

**Kernel dynamics.**   Kernel methods (Hofmann et al., 2008) have been classic machine learning techniques, where the primary goal is to design an effective embedding that maps inputs to a feature space, thus facilitating efficient algorithms to find good solutions (e.g., linear classifier). While neural networks are inherently complex, seminal works (Jacot et al., 2018; Lee et al., 2019) have shown that in the infinite width limit, a network can be linearized by its *neural tangent kernel (NTK)*. Thus, studying the NTK of a network allows an analytical understanding of various properties of neural networks, such as convergence to global minima (Du et al., 2018; 2019), generalization performance (Allen-Zhu et al., 2019; Arora et al., 2019), implicit bias (Bordelon et al., 2020; Canatar et al., 2021), and neural scaling laws (Bahri et al., 2024).

When a network is properly initialized (Chizat et al., 2019), gradient descent can converge to the NTK of the random initialization, a setting known as the *kernel regime* (a.k.a., *lazy training* or *random feature regime*). On the other hand, a network can also enter what is known as the *feature learning regime* (a.k.a., *rich training* or *mean-field limit*), where it deviates from the NTK of the initialization (Geiger et al., 2020). Extensive research has been conducted to characterize lazy versus rich regimes (Geiger et al., 2020; Woodworth et al., 2020) and to demonstrate instances where feature learning outperforms lazy training (Yang & Hu, 2021; Ba et al., 2022; Dandi et al., 2023). It is important to note that even when a network undergoes feature learning, the NTK can still be defined at each epoch. Previous works also analytically characterized the dynamics of kernel in simpler models (Bordelon et al., 2020). Studying such kernel dynamics also provides a lens for exploring questions related to feature learning, such as grokking (Kumar et al., 2024).

**Representational geometry.**   The visualization approaches mentioned above focus on studying the geometric properties of the feature map itself. Another fruitful direction is to examine the geometric properties of the neural representations of inputs (i.e., embedding vectors) and their connections to performance (Chung & Abbott, 2021; Gurnani & Gajic, 2023). Various dimensionality reduction methods (e.g., principal components analysis (PCA), Isomap, t-SNE, MDS, and UMAP) have been proposed to build intuitions about the organization of high-dimensional feature spaces. In addition, there are approaches that study lower-order statistics of embedding vectors, such as representational similarity (Kriegeskorte & Kievit, 2013) and spectral methods (Rahaman et al., 2019; Bahri et al., 2024; Ghosh et al., 2022). Methods for extracting higher-level geometric properties (e.g., dimension) have also been proposed (Chung et al., 2018; Cohen et al., 2020; Chou et al., 2025; Ansuini et al., 2019), with wide applications in both machine learning (e.g., memorization (Stephenson et al., 2021), grokking of modular arithmetic (Liu et al., 2022; Nanda et al., 2023), in-context learning in LLM (Kirsanov et al., 2025), self-supervised learning (Yerxa et al., 2023; Kuoch et al., 2024)), and neuroscience (e.g., perceptual untangling in object categorization (Chung et al., 2018), olfactory memory (Hu et al., 2024), abstraction (Bernardi et al., 2020), few-shot learning (Sorscher et al., 2022), social learning (Paraouty et al., 2023)).

## A.1. Previous work on storage capacity

Storage capacity is defined as the information load for linear readouts and has been studied in several communities, including learning theory (Cover, 1965) and statistical physics of neural networks (Gardner & Derrida, 1988; Gardner, 1988). To enable a mathematical treatment, we focus on the proportional limit (a.k.a. the high-dimensional limit, the thermodynamic limit), i.e., $N, P \to \infty$ and $\lim_{N,P \to \infty} N/P = O(1)$. For a given network and input data, we denote the representation of

the $i$-th input $x_i$ as $\Phi(x_i) \in \mathbb{R}^N$ where $\Phi$ is the (non-linear) feature map. The storage capacity of $\Phi$ is defined as.

$$\alpha(\Phi) := \lim_{N \to \infty} \max_P \left\{ \frac{P}{N} \; : \; \Pr_{\mathbf{y}} \left[ \exists \theta \in \mathbb{R}^N, \; \forall i \in [P], \; y_i \langle \theta, \Phi(x_i) \rangle \geq 0 \right] \geq 1 - o_N(1) \right\} \tag{2}$$

where $\mathbf{y} \in \{\pm 1\}^P$ is uniformly random sampled, $\theta$ is the linear classifier, and $o_N(1)$ denotes vanishing terms (i.e., $o_N(1) \to 0$ as $N \to \infty$). One can also consider the setting where the distribution of $\mathbf{y}$ is biased toward some task direction (Montanari et al., 2025). Intuitively, $\alpha(\Phi)$ quantifies the number of patterns per neuron that a network can store and decode with linear readouts.

Recall that storage capacity is defined as the critical ratio between the number of stored patterns and the number of neurons (Equation (2)). Cover's theorem (Cover, 1965) shows that the success probability of having a linear classifier for $P$ points with random binary labels in general position [8] is $p(N, P) = 2^{1-P} \sum_{k=0}^{N-1} \binom{P-1}{k}$. In particular, for $P/N < 2$ we have $\lim_{N \to \infty} p(N, P) = 0$ and for $P/N > 2$ we have $\lim_{N \to \infty} p(N, P) = 1$. Namely, the storage capacity of points in general position with random binary label is 2. See also Figure 8 for finite-size and numerical examples.

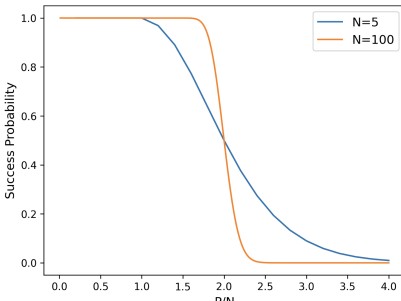 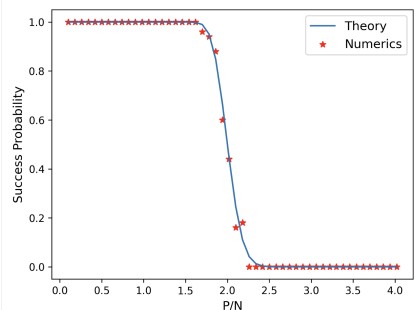

*Figure 8.* Storage capacity of random points and labels. Storage capacity is defined as the critical ration $P/N = 2$ where the success probability undergoes a phase transition. Left: finite size success probability curves proved in Cover's theorem. Right: a numerical check for Cover's theorem.

In the seminal works of Gardner and Derrida (Gardner & Derrida, 1988; Gardner, 1988), the storage capacity for random points with non-zero margin is analytically characterized using replica method. In the context of associative memory, the storage capacity of Hopfield networks (Hopfield, 1982) is calculated by (Amit et al., 1987).

## B. Manifold Capacity Theory and Effective Geometry

Manifold capacity theory (MCT)(Chung et al., 2018; Chung & Abbott, 2021; Wakhloo et al., 2023; Chou et al., 2025) was originally developed for the study of manifold untangling (DiCarlo & Cox, 2007) in theoretical/computational neuroscience. Intuitively, manifold untangling refers to the increased separation of high-dimensional manifolds (e.g., point cloud manifolds) in the eyes of a downstream readout. MCT quantifies this intuition via modeling a downstream neuron as a linear classifier, and uses the *packing efficiency* of the neural representational space to evaluate the degree of manifold untangling. Mathematically, such packing efficiency coincides with support vector machine (SVM) in an average-case setting.

### B.1. Neural manifolds as convex hulls of pre-readout representations

As we are studying feature learning, we are interested in the neural representations that correspond to activations obtained from the pre-linear readout layer neurons. The readers can refer to Appendix D and Appendix E for details on activation extraction. Notation wise, let $N$ be the number of neurons. Therefore, all neural representations live in $\mathbb{R}^N$ space.

Next, we group neural representations by their category labels assigned during training to obtain $P$ data manifolds. For $i \in \{1, \ldots, P\}$, the $i$-th data manifold, denoted as $\mathcal{M}_i$, is a convex set in $\mathbb{R}^N$. To ensure convexity in practice, we take $M_i$ to be the convex hull of a collection of vectors $\mathcal{M}_i = \{\mathbf{x}_1^i, \ldots, \mathbf{x}_{M_i}^i\}$ where $M_i$ is the number of points in the $i$-th manifold.

---

[8]Meaning that every $N' \leq N$ points are linearly independent. Note that random points are in general position with probability $1 - o(1)$.

Notice that the each data manifold lives in its own subspace of dimension $D_i \leq N$. Therefore, we can rewrite each data manifold in its own coordinate system:

$$\mathcal{M}_i = \left\{ \mathbf{u}_0^i + \sum_{j=1}^{D_i} s_j \mathbf{u}_j^i \,\middle|\, \mathbf{s} = (s_1, \ldots, s_{D_i}) \in \mathcal{S}_i \right\} \tag{3}$$

Here, $\mathbf{u}_0^i$ is the center of the $i$-th manifold and $\{\mathbf{u}_j^i\}_{j=1}^{D_i}$ is an orthonormal basis. The shape set $\mathcal{S}_i \subset \mathbb{R}^{D_i}$ is a convex set denoting coordinates of the manifold points in its subspace. In practice, the manifold axes and shape sets $\mathcal{S}_i$ are completely data driven.

## B.2. A simulation definition for manifold capacity

Recall from Section 2 that the simulation version of manifold capacity is defined as follows.

**Definition B.1** (Simulated manifold capacity (Chou et al., 2025)). Let $P, N \in \mathbb{N}$ and $\mathcal{M}_i \subseteq \mathbb{R}^N$ be convex sets for each $i \in [P] = \{1, \ldots, P\}$. For each $n \in [N]$, define

$$p_n := \Pr_{\mathbf{y}, \Pi_n} \left[ \exists \theta \in \mathbb{R}^n \,:\, y_i \langle \theta, \mathbf{s} \rangle \geq 0, \, \forall i \in [P], \, \mathbf{s} \in \mathcal{M}_i \right]$$

where $\mathbf{y}$ is a dichotomy vector randomly sampled[9] from $\mathcal{Y} \subset \{\pm 1\}^P$ and $\Pi_n$ is a random projection from $\mathbb{R}^N$ to $\mathbb{R}^n$. Suppose $p_N = 1$ (which means that the input manifolds are linearly separable in $\mathbb{R}^N$), the simulated capacity of $\{\mathcal{M}_i\}_{i \in [P]}$ is defined as

$$\alpha_{\mathsf{sim}} := \frac{P}{\sum_n (1 - p_n)}.$$

Intuitively, the simulated manifold capacity measures the *packability* (Chung et al., 2018) of manifolds by determining the smallest dimensional subspace needed to ensure they can be separated. Namely, manifolds that are more packable[10] (i.e., separable when projected to smaller dimensional subspaces) exhibit higher manifold capacity. Note that the simulated capacity can be estimated from data by empirically estimate $p_n$ and perform binary search to find the critical dimension $\min_{p_n \geq 0.5} \{n\}$. This procedure is computationally expensive and requires some choices of hyperparameters (which makes the definition a little ad hoc). Nevertheless, Definition 2.1 provides good intuition on how to think about manifold capacity (and its connection to packing).

We remark that one don't necessarily need to sample $\mathbf{y}$ uniformly at random from $\{\pm 1\}^P$. It is also reasonable to fix a choice of $\mathbf{y}$ or sample $\mathbf{y}$ from a subset of $\{\pm 1\}^P$ (e.g., all the 1-versus-rest dichotomies) as discussed in (Chou et al., 2025).

## B.3. A mean-field definition for manifold capacity

To overcome the above-mentioned drawbacks of simulated manifold capacity, previous work (Chung et al., 2018; Wakhloo et al., 2023; Chou et al., 2025) defined a *mean-field models* to enable a mathematical definition of manifold capacity while providing a good approximation to the simulated manifold capacity. In particular, in this paper we use the GLUE (Geometry Linked to Untangling Efficiency) theory developed in (Chou et al., 2025).

**Mean-field model from GLUE (Chou et al., 2025).** Given a collection of (finite) data manifolds $\{\mathcal{M}_i\}_{\mu=1}^P$. A mean-field model is to generate infinitely many ($P_M$) manifolds in an infinite-dimensional ($N_M$) space and characterizing the largest possible $P_M/N_M$ such that these "mean-field" manifolds are separable. The key idea is that if this generating process nicely preserve the structure in the data manifolds, then the packing property of these mean-field manifolds will be very similar

**Definition B.2** (Mean-field model from (Chou et al., 2025)). Let $\{\mathcal{M}_i\}_{i \in [P]}$ be a collection of data manifolds in $\mathbb{R}^N$ as defined in Equation 3. Let $\alpha \in \mathbb{R}_{\geq 0}$ and $P_M, N_M$ be integers with the following properties: (i) $P_M, N_M \to \infty$ and (ii) $P_M/N_M = \alpha < \infty$, and $P_M$ be divisible by $P$. We define the mean-field manifolds $\mathcal{M}_M(P_M, N_M) = \{\mathcal{M}_M^{a,i}\}_{a \in [P_M/P], i \in [P]}$ as follows.

---

[9]Typically we pick $\mathcal{Y}$ to be the collection of all 1-versus-rest dichotomies.

[10]The reason why this is called "packing" is that projecting manifolds into smaller dimensional subspace is like packing them into a smaller neural representational space.

---

**Algorithm 1** Estimate simulated manifold capacity

---

**Input:** $\{\mathcal{M}_i\}$: $P$ point clouds, each containing $M$ points in an $N$-dimensional ambient space.
**Output:** $\alpha_{\mathsf{sim}}$: Simulated manifold capacity.

   $n^* \leftarrow \text{BinarySearch}(\{\mathcal{M}_i\}, 1, N)$.
   $\alpha_{\mathsf{sim}} \leftarrow P/n^*$.
   **return** $\alpha_{\mathsf{sim}}$.

   % Binary search for the smallest $n$ such that $p_n \geq 0.5$.
   **function** BinarySearch($\{\mathcal{M}_i\}, n_l, n_r$)
      $n_m \leftarrow \lfloor (n_l + n_r)/2 \rfloor$.
      **if** $n_m = n_l$ **then**
         **return** $n_m$.
      **else**
         $p_{n_m} \leftarrow \text{EstProb}(\{\mathcal{M}_i\}, n_m)$.
         **if** $p_{n_m} > 0.5$ **then**
            **return** BinarySearch($\{\mathcal{M}_i\}, n_l, n_m$).
         **else**
            **return** BinarySearch($\{\mathcal{M}_i\}, n_m, n_r$).
         **end if**
      **end if**
   **end function**

   % Estimate the probability of linear separability after random projection.
   **function** EstProb($\{\mathcal{M}_i\}, n, m = 1000$)
      $\mathsf{cnt} \leftarrow 0$
      **for** $i$ from 1 to $m$ **do**
         $\Pi \leftarrow$ a random projection from $\mathbb{R}^N$ to $\mathbb{R}^n$.
         $\mathbf{y} \leftarrow$ a random vector from $\{\pm 1\}^P$.
         $\{\mathcal{M}_i'\} \leftarrow \{\Pi \mathcal{M}_i\}$.
         **if** $\{\mathcal{M}_i'\}$ is linearly separable **then**
            $\mathsf{cnt} \leftarrow \mathsf{cnt} + 1$.
         **end if**
      **end for**
      **return** $\mathsf{cnt}/m$.
   **end function**

---

- First, find an orthogonal basis $\{\mathbf{e}_k\}_{k=1}^N$ in $\mathbb{R}^N$ for the basis vectors of all the data manifolds. Namely, for each $i \in [P]$, there exists a linear transformation $Q^i \in \mathbb{R}^{(D_i+1) \times N}$ such that $\mathbf{u}_j^i = \sum_k Q_k^{i,j} \mathbf{e}_k$ for each $j \in \{0, 1, \ldots, D_i\}$.

- Next, for each $a \in [P_M/P]$, generate $\mathbf{v}_1^a, \ldots, \mathbf{v}_N^a \sim \mathcal{N}(0, I_{N_M})$ independently and let $\mathbf{V}^a$ be the $N_M \times N$ matrix with $\mathbf{v}_j^a$ on its columns.

- Define $M_M^{a,i} = \left\{ (\mathbf{V}^a Q^i)_0 + \sum_{j=1}^{D_i} s_j (\mathbf{V}^a Q^i)_j \ : \ \mathbf{s} = (s_1, \ldots, s_{D_i}) \in \mathcal{S}_i \right\}$ as the $i$-th manifold in the $a$-th cloud where $(\mathbf{V}^a Q^i)_i = \sum_k \mathbf{v}_k^a Q_k^{i,j}$ for every $a \in [P_M/P]$ and $i \in [P]$.

Now, we are ready to formally define the mean-field version of manifold capacity.

**Definition B.3** (Mean-field manifold capacity (Chung et al., 2018; Chou et al., 2025))**.** Let $\{\mathcal{M}_i\}_{i \in [P]}$ be a collection of data manifolds in $\mathbb{R}^N$ as defined in Equation 3. The manifold capacity of $\{\mathcal{M}_i\}_{i \in [P]}$ is defined as

$$\alpha_M := \lim_{N_M \to \infty} \max_{P_M} \left\{ \frac{P_M}{N_M} \ : \ \Pr_{\mathbf{y}, \mathcal{M}_M(P_M, N_M)} \left[ \begin{array}{c} \exists \theta \in \mathbb{R}^{N_M}, \ \forall a \in [P_M/P], \ i \in [P], \\ \min_{\mathbf{s} \in \mathcal{M}_M^{a,i}} y_i \langle \theta, \mathbf{s} \rangle \geq 0 \end{array} \right] \geq 1 - o_{N_M}(1) \right\}$$

where and $o_{N_M}(1) \to 0$ as $N_M \to \infty$.

Finally, previous work (Chung et al., 2018; Chou et al., 2025) derived a formula for mean-field manifold capacity as follows.

$$\alpha_M^{-1} = \frac{1}{P} \mathop{\mathbb{E}}_{\substack{\mathbf{y} \sim \{\pm 1\}^P \\ T \sim \mathcal{N}(0, I_N)}} \left[ \max_{\mathbf{s}_i \in \mathcal{M}_i} \left\{ \| \mathsf{proj}_{\mathsf{cone}(\{y_i \mathbf{s}_i\})} \mathbf{t} \|_2^2 \right\} \right] \qquad (4)$$

$$= \frac{1}{P} \mathop{\mathbb{E}}_{\substack{\mathbf{y} \sim \{\pm 1\}^P \\ T \sim \mathcal{N}(0, I_N)}} \left[ \max_{\substack{\mathbf{s}_i \in \mathcal{M}_i \\ \lambda_i \geq 0}} \left\{ \left( \frac{-T \cdot \sum_i \lambda_i y_i \mathbf{s}_i}{\| \sum_i \lambda_i y_i \mathbf{s}_i \|_2} \right)_+^2 \right\} \right]$$

where $\mathcal{N}(\mu, \Sigma)$ denotes the multivariate Gaussian distribution with mean $\mu$ and covariance $\Sigma$ and $\mathsf{cone}(\cdot)$ is the convex cone spanned by the vectors, i.e., $\mathsf{cone}(\{y_i \mathbf{s}_i\}) = \{\sum_i \lambda_i y_i \mathbf{s}_i \ : \ \lambda_i \geq 0\}$.

### B.4. Effective geometric measures from capacity formula

The advantages of mean-field manifold capacity are: (i) $\alpha_M$ can be estimated via solving a quadratic program (Algorithm 2) and (ii) Equation 1 connects manifold capacity to the structure of the manifolds $\{\mathcal{M}_i\}$. Specifically, for each $\mathbf{y}, \mathbf{t}$, define $\{\mathbf{s}_i(\mathbf{y}, \mathbf{t})\} = y_i \cdot \arg \max_{\{\mathbf{s}_i\}} \| \mathsf{proj}_{\mathsf{cone}(\{y_i \mathbf{s}_i\})} \mathbf{t} \|_2^2$ as the *anchor points* with respect to $\mathbf{y}$ and $T$. Intuitively, these anchor points are the support vectors with respect to some random projection and dichotomy as in Definition 2.1. Specifically, these anchor points are analytically linked to manifold capacity via Equation 1 and are distributed over the manifolds $\{\mathcal{M}_i\}$. This connection inspired the previous work (Chung et al., 2018; Chou et al., 2025) to define the following effective manifold geometric measures that capture the structure of manifolds while being analytically connected to capacity.

The first key idea of defining effective geometric measure is the segregation of anchor points into their *center part* and their *axis part*. Concretely, for each $i \in [P]$, define $\mathbf{s}_i^0 := \mathbb{E}_{\mathbf{y}, \mathbf{t}}[\mathbf{s}_i(\mathbf{y}, \mathbf{t})]$ as the center of the $i$-th manifold and define $\mathbf{s}_i^1(\mathbf{y}, \mathbf{t}) := \mathbf{s}_i(\mathbf{y}, \mathbf{t}) - \mathbf{s}_i^0$ to be the axis part of $\mathbf{s}_i(\mathbf{y}, \mathbf{t})$ for each pair of $(\mathbf{y}, \mathbf{t})$.

Next, (Chung et al., 2018) used an identity: $a = \frac{b}{1 + \frac{b-a}{a}}$, and set $a = \| \mathsf{proj}_{\mathsf{cone}(\{\mathbf{s}_i(\mathbf{y}, \mathbf{t})\}_i)} \mathbf{t} \|_2^2$ and $b = \| \mathsf{proj}_{\mathsf{cone}(\{\mathbf{s}_i^1(\mathbf{y}, \mathbf{t})\}_i)} \mathbf{t} \|_2^2$ to rewrite the capacity formula (Equation 4) as follows.

$$\alpha_M^{-1} = \frac{1}{P} \mathop{\mathbb{E}}_{\mathbf{y}, \mathbf{t}} \left[ \| \mathsf{proj}_{\mathsf{cone}(\{\mathbf{s}_i(\mathbf{y}, \mathbf{t})\}_i)} \mathbf{t} \|_2^2 \right]$$

$$= \frac{1}{P} \mathop{\mathbb{E}}_{\mathbf{y}, \mathbf{t}} \left[ \frac{\| \mathsf{proj}_{\mathsf{cone}(\{\mathbf{s}_i^1(\mathbf{y}, \mathbf{t})\}_i)} \mathbf{t} \|_2^2}{1 + \frac{\| \mathsf{proj}_{\mathsf{cone}(\{\mathbf{s}_i^1(\mathbf{y}, \mathbf{t})\}_i)} \mathbf{t} \|_2^2 - \| \mathsf{proj}_{\mathsf{cone}(\{\mathbf{s}_i(\mathbf{y}, \mathbf{t})\}_i)} \mathbf{t} \|_2^2}{\| \mathsf{proj}_{\mathsf{cone}(\{\mathbf{s}_i(\mathbf{y}, \mathbf{t})\}_i)} \mathbf{t} \|_2^2}} \right] .$$

Then, they proceeded with the following approximation.

$$\approx \frac{\frac{1}{P}\,\mathbb{E}_{\mathbf{y},\mathbf{t}}\left[\|\mathsf{proj}_{\mathsf{cone}(\{\mathbf{s}_i^1(\mathbf{y},\mathbf{t})\}_i)}\mathbf{t}\|_2^2\right]}{\mathbb{E}_{\mathbf{y},\mathbf{t}}\left[1 + \frac{\|\mathsf{proj}_{\mathsf{cone}(\{\mathbf{s}_i^1(\mathbf{y},\mathbf{t})\}_i)}\mathbf{t}\|_2^2 - \|\mathsf{proj}_{\mathsf{cone}(\{\mathbf{s}_i(\mathbf{y},\mathbf{t})\}_i)}\mathbf{t}\|_2^2}{\|\mathsf{proj}_{\mathsf{cone}(\{\mathbf{s}_i(\mathbf{y},\mathbf{t})\}_i)}\mathbf{t}\|_2^2}\right]} \; . \tag{5}$$

(Chung et al., 2018; Chou et al., 2025) found that the above approximation empirically performs well. Furthermore, as the numerator mimics the notion of Gaussian width of a convex body and the denominator behaves like (normalized) radius of a sphere, they defined effective manifold dimension and radius as follows.

**Definition B.4** (Effective manifold geometric measures (Chung et al., 2018; Chou et al., 2025))**.** For each $i \in [P]$, define $\mathbf{s}_i^0 := \mathbb{E}_{\mathbf{y},\mathbf{t}}[\mathbf{s}_i(\mathbf{y},\mathbf{t})]$ as the **center** of the $i$-th manifold and define $\mathbf{s}_i^1(\mathbf{y},\mathbf{t}) := \mathbf{s}_i(\mathbf{y},\mathbf{t}) - \mathbf{s}_i^0$ to be the **axis** part of $\mathbf{s}_i(\mathbf{y},\mathbf{t})$ for each pair of $(\mathbf{y},\mathbf{t})$.

- **Manifold dimension** captures the degree of freedom of the noises/variations within the manifolds. Formally, it is defined as $D_M := \mathbb{E}_{\mathbf{y},\mathbf{t}}[\|\mathsf{proj}_{\mathsf{cone}(\{\mathbf{s}_i^1(\mathbf{y},\mathbf{t})\}_i)}\mathbf{t}\|_2^2]$.

- **Manifold radius** captures the noise-to-signal ratio of the manifolds. Formally, it is defiend as $R_M :=$
$\sqrt{\mathbb{E}_{\mathbf{y},\mathbf{t}}\left[\frac{\|\mathsf{proj}_{\mathsf{cone}(\{\mathbf{s}_i(\mathbf{y},\mathbf{t})\}_i)}\mathbf{t}\|^2}{\|\mathsf{proj}_{\mathsf{cone}(\{\mathbf{s}_i^1(\mathbf{y},\mathbf{t})\}_i)}\mathbf{t}\|^2 - \|\mathsf{proj}_{\mathsf{cone}(\{\mathbf{s}_i(\mathbf{y},\mathbf{t})\}_i)}\mathbf{t}\|^2}\right]}.$

- **Center alignment** captures the correlation between the center of different manifolds. Formally, it is defined as $\rho_M^c := \frac{1}{P(P-1)}\sum_{i\neq j}|\langle\mathbf{s}_i^0,\mathbf{s}_j^0\rangle|$.

- **Axis alignment** captures the correlation between the axis of different manifolds. Formally, it is defined as $\rho_M^a := \frac{1}{P(P-1)}\sum_{i\neq j}\mathbb{E}_{\mathbf{y},\mathbf{t}}[|\langle\mathbf{s}_i^1(\mathbf{y},\mathbf{t}),\mathbf{s}_j^1(\mathbf{y},\mathbf{t})\rangle|]$.

- **Center-axis alignment** captures the correlation between the center and axis of different manifolds. Formally, it is defined as $\psi_M := \frac{1}{P(P-1)}\sum_{i\neq j}\mathbb{E}_{\mathbf{y},\mathbf{t}}[|\langle\mathbf{s}_i^0,\mathbf{s}_j^1(\mathbf{y},\mathbf{t})\rangle|]$.

**A capacity approximation formula by dimension and radius.** Recall that in Equation 5 previous work (Chung et al., 2018) used the identity $a = \frac{b}{1+\frac{b-a}{a}}$ to approximate the manifold capacity. After defining manifold dimension and radius, one can then plug them back to Equation 5 and get the following approximation of manifold capacity via effective manifold dimension and radius.

$$\alpha_M \approx \frac{1 + R_M^{-2}}{D_M} \; . \tag{6}$$

### B.5. Connections between manifold capacity and its effective geometric measures

Here, we demonstrate the connections between manifold capacity and its effective geometric measures by synthetic manifolds. In particular, we consider isotropic Gaussian clouds parametrized by a set of *ground truth* latent parameters: dimension $D_{\mathrm{ground}}$, radius $R_{\mathrm{ground}}$, center correlations $\rho^c\mathrm{ground}$, axis correlations $\rho^a_{\mathrm{ground}}$, and center-axis correlations $\psi_{\mathrm{ground}}$. See Section D.1.1 for more details on the generative process. In this section, we focus on showing that the effective geometric measures $D_M, R_M, \rho_M^c, \rho_M^a, \psi_M$ capture the corresponding ground truth parameter.

**Effective manifold dimension and radius.** We first set all the manifold correlations to be zero and vary the ground truth radius and dimension. Here we pick $N = 1000$ neurons, $P = 2$ manifold, $M = 200$ points per manifold, varying the underlying dimension from 2 to 10, and varying the underlying radius from 0.8 to 2. In Figure 9, we vary the ground truth dimension in the x-axis, and in Figure 10, we vary the ground truth radius in the x-axis.

**Effective alignment measures.** Next, we fix the ground truth dimension to be $D_{\mathrm{ground}} = 4$ and radius to be $R_{\mathrm{ground}} = 1$ and vary $\rho_{\mathrm{ground}}^c, \rho_{\mathrm{ground}}^a, \psi_{\mathrm{ground}}$ from 0 to 0.8. In Figure 11, we vary the center correlations, and in Figure 12, we vary the axis correlations.

### B.6. Algorithms for estimating manifold capacity and effective geometric measure

We provide pseudocodes for estimating manifold capacity and effective geometric measure in Algorithm 2.

---

**Algorithm 2** Estimate manifold capacity and effective geometric measures

---

**Input:** $\{\mathcal{M}_i\}$: $P$ point clouds, each containing $M$ points in an $N$-dimensional ambient space; $n_t$: number of samples for estimating the expectation.

**Output:** $\alpha_M$: Manifold capacity; $D_M$: Effective dimension; $R_M$: Effective radius; $\rho_M^a$: Effective axis alignment; $\rho_M^c$: Effective center alignment; $\psi_M$: Effective center-axis alignment.

  % Step 1: Sample anchor points.
  **for** $k$ from 1 to $n_t$ **do**
    $\mathbf{t}_k \leftarrow$ a vector sampled from isotropic $N$-dimensional Gaussian distribution.
    $\mathbf{y} \leftarrow$ a random dichotomy vector from $\{\pm 1\}^P$.
    $\mathbf{A} \leftarrow I_N; \mathbf{q} \leftarrow -\mathbf{t}_k; \mathbf{h} \leftarrow \mathbf{0}_N$.
    $\mathbf{G} \leftarrow (y \odot \{\mathcal{M}_i\}_{i=1}^P)$. $\{\mathbf{G}_{i,j} = y_i\mathbf{s}$ is a row vector where $\mathbf{s}$ is the $j$-th point in $\mathcal{M}_i$.$\}$
    output $\leftarrow qp(\mathbf{A}, \mathbf{q}, \mathbf{G}, \mathbf{h})$. $\{\min_{\mathbf{x}} \frac{1}{2}\mathbf{x}^\top \mathbf{A}\mathbf{x} + \mathbf{q}^\top \mathbf{x}$ s.t. $\mathbf{G}\mathbf{x} \leq \mathbf{h}$. $\}$
    $\mathbf{z}_{\text{dual}} \leftarrow$ output["dual"] $\{$The support vectors $\}$
    **for** $i$ from 1 to $P$ **do**
      $\mathbf{s}_i[k] \leftarrow \sum_j (\mathbf{z}_{\text{dual}})_{i,j}^\top \mathbf{G} / \sum_j (\mathbf{z}_{\text{dual}})_{i,j}$
    **end for**
  **end for**

  % Step 2: Estimate (anchor) manifold centers.
  **for** $i$ from 1 to $P$ **do**
    $\mathbf{s}_i^0 \leftarrow \frac{1}{n_t} \sum_{k=1}^{n_t} \mathbf{s}_i[k])$.
  **end for**
  $\mathbf{G}^0 \leftarrow \sum_i \mathbf{s}_i^0 (\mathbf{s}_i^0)^\top$. $\{$Anchor center gram matrix.$\}$

  % Step 3: Separate the center and axis part of anchor points.
  **for** $k$ from 1 to $n_t$ **do**
    **for** $i$ from 1 to $P$ **do**
      $\mathbf{s}_i^1[k] \leftarrow \mathbf{s}_i[k] - \mathbf{s}_i^0$. $\{$The axis part of the anchor poitn in the $i$-th manifold.$\}$
    **end for**
    $\mathbf{t}^1[k] \leftarrow \sum_i \mathbf{s}_i^1[k]\mathbf{t}_k$.
    $\mathbf{G}^1[k] \leftarrow \sum_i \mathbf{s}_i^1[k](\mathbf{s}_i^1[k])^\top$. $\{$Anchor axis gram matrix.$\}$
  **end for**

  % Step 4: Estimate manifold capacity and effective geometric measures.
  $\alpha_M \leftarrow (\frac{1}{n_t P} \sum_{k=1}^{n_t} (\mathbf{s}_i[k]\mathbf{t}_k)^\top (\mathbf{s}_i[k](\mathbf{s}_i[k]^\top)^\dagger (\mathbf{s}_i[k]\mathbf{t}_k))^{-1}$.
  $D_M \leftarrow \frac{1}{n_t P} \sum_{k=1}^{n_t} \mathbf{t}^1[k]^\top \mathbf{G}^1[k]^\dagger \mathbf{t}^1[k]$.
  $R_M \leftarrow \sqrt{\frac{1}{n_t} \sum_{k=1}^{n_t} \frac{\mathbf{t}^1[k]^\top (\mathbf{G}^1[k]+\mathbf{G}^0)^\dagger \mathbf{t}^1[k]}{\mathbf{t}^1[k]^\top (\mathbf{G}^1[k]+\mathbf{G}^1[k](\mathbf{G}^0)^\dagger \mathbf{G}^1[k])^\dagger \mathbf{t}^1[k]}}$. $\{$Equivalent to the definition of radius after applying the Woodbury formula for numerical stabiltiy.$\}$
  $\rho_M^c \leftarrow \frac{1}{P(P-1)} \sum_{i=1}^P \sum_{i \neq j} \frac{(\mathbf{s}_i^0)^\top \mathbf{s}_j^0}{\|\mathbf{s}_i^0\|_2 \cdot \|\mathbf{s}_j^0\|_2}$.
  $\rho_M^a \leftarrow \frac{1}{P(P-1)} \sum_{i=1}^P \sum_{j \neq i} \frac{1}{n_k} \sum_{k=1}^{n_k} \frac{\mathbf{s}_i^1[k]^\top \mathbf{s}_j^1[k]}{\|\mathbf{s}_i^1[k]\|_2 \cdot \|\mathbf{s}_j^1[k]\|_2}$.
  $\psi_M \leftarrow \frac{1}{P(P-1)} \sum_{i=1}^P \sum_{j \neq i} \frac{1}{n_k} \sum_{k=1}^{n_k} \frac{(\mathbf{s}_i^0)^\top \mathbf{s}_j^1[k]}{\|\mathbf{s}_i^0\|_2 \cdot \|\mathbf{s}_j^1[k]\|_2}$.
  **return** $\alpha_M, D_M, R_M, \rho_M^a, \rho_M^c, \psi_M$.

---

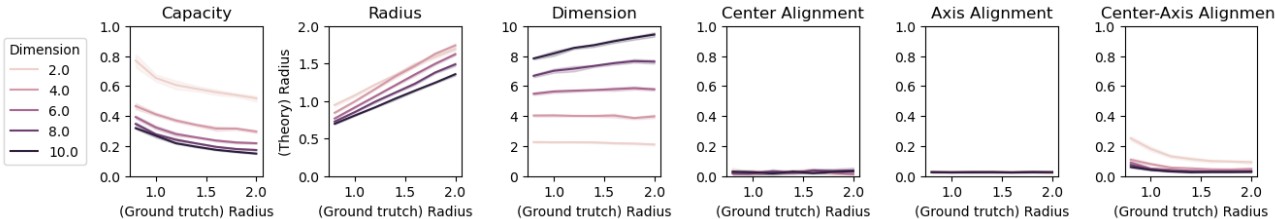

*Figure 9.* Effective manifold dimension tracks the ground truth dimension of uncorrelated isotropic Gaussian clouds. Note that the higher the dimension, the smaller capacity, as discussed in Figure 2c.

*Figure 10.* Effective manifold radius tracks the ground truth radius of uncorrelated isotropic Gaussian clouds. Note that the higher the radius, the smaller capacity, as discussed in Figure 2c.

## C. Theoretical Results

We consider the training of a fully-connected 2-layer network of the form $f(\mathbf{x}) = \frac{1}{\sqrt{N}}\mathbf{a}^\top \sigma(W^\top \mathbf{x})$, where $\mathbf{x} \in \mathbb{R}^d$ is an input, $W \in \mathbb{R}^{N \times d}$ is the hidden layer matrix, $\mathbf{a} \in \mathbb{R}^N$ is the readout weight, and $\sigma : \mathbb{R} \to \mathbb{R}$ is the (non-linear) activation function. To study feature learning in this setting, it is common to consider $W$ to be randomly initialized (i.e., random feature model (Rahimi & Recht, 2007)) and update via gradient descent with squared loss. Meanwhile, the readout weight $\mathbf{a}$ is randomly initialized and fixed to avoid lazy learning (where the network minimally adjusts the hidden layer and focuses on learning a good readout weight) as well as enable mathematical analysis (Ba et al., 2022). Input data and label $(\mathbf{x}_1, y_1), \ldots, (\mathbf{x}_{P_{\text{train}}}, y_{P_{\text{train}}})$ were randomly generated by a teacher-student setting, where there is a hidden signal direction $\beta^*$ that correlates with the label (see Setting C.2 for the full setting). As previously proved in (Ba et al., 2022) (see Proposition C.5), in the proportional asymptotic limit (i.e., $P_{\text{train}}, d, N \to \infty$ at the same rate), the first-step gradient update can be approximated by a rank-1 matrix that contains label information, resulting in the updated weight to be more aligned with the hidden signal $\beta^*$. Hence, in this setting, the learning rate $\eta$ can be used as the ground-truth to measure the amount of task-relevant information (i.e., richness in learning) in the model representation after gradient updates.

We extend the previous results in (Ba et al., 2022) from a regression setting to a classification setting. Specifically, We prove that capacity correctly tracks the effective degree of richness after one gradient step[11]. Moreover, we derive a monotone connection between capacity and prediction accuracy, thereby justifying the use of capacity as a measure of richness in feature learning within a well-studied theoretical setting. Here, we provide an informal statement of our results and leave the formal version and proof in Appendix C.

### C.1. Formal theorem statement

Let $d \in \mathbb{N}$ be the input dimension and $N \in \mathbb{N}$ be the number of hidden units. Let $W_0 \in \mathbb{R}^{N \times d}$ be the weight matrix of a fully connected 2-layer neural network. The feature of an input vector is defined as $\Phi_0(\mathbf{x}) = \sigma(W_0\mathbf{x})$ where $\sigma(\cdot) : \mathbb{R} \to \mathbb{R}$ is a non-linear activation function, e.g., ReLU or tanh. The readout weight is denoted as $\mathbf{a} \in \mathbb{R}^N$. Finally, the output of the 2-layer NN is the sign of the readout, i.e., $f(\mathbf{x}) = \mathsf{sgn}(\mathbf{a}^\top \Phi(\mathbf{x}))$.

---

[11]Here we follow the convention in (Ba et al., 2022) and study only the first gradient step as the key Gaussian equivalence step might not hold for more steps as remarked in footnote 2 of (Ba et al., 2022).

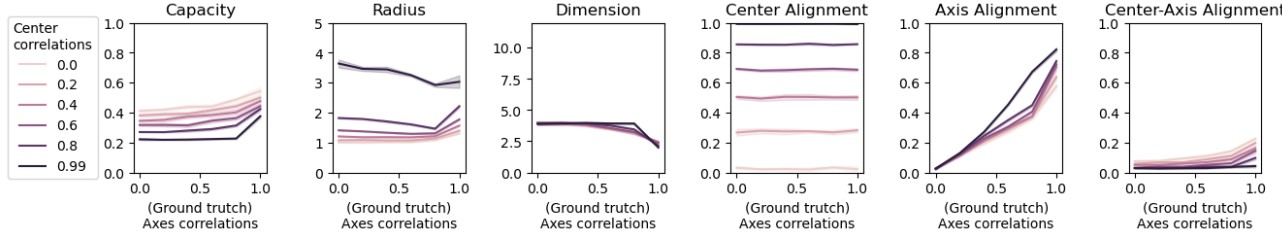

*Figure 11.* Effective manifold center alignment tracks the ground truth center correlations of isotropic Gaussian clouds. Note that the higher the center alignment, the smaller capacity, as discussed in Figure 2c. Also, in the large center correlations regime, the effective radius increases.

*Figure 12.* Effective manifold axis alignment tracks the ground truth axis correlations of isotropic Gaussian clouds. Note that the higher the axis alignment, the higher capacity, as discussed in Figure 2c. Also, in the large axis correlations regime, the effective dimension decreases.

Let $\{(\mathbf{x}_i, y_i)\}_{i \in [P_{\text{train}}]}$ be the collection of training data. We consider gradient descent over the mean square error (MSE) of the 2-layer NN, i.e., $\mathcal{L}(f) = \frac{1}{P_{\text{train}}} \sum_{i \in [P_{\text{train}}]} \ell(f(\mathbf{x}_i), y_i)$ where $\ell(z_i, y_i) = \frac{1}{2}(z - y)^2$. The gradient update with learning rate $\eta > 0$ is $W_{t+1} = W_t + \eta G_t$ where

$$G_t = \frac{1}{P_{\text{train}}} \sum_{i \in [P_{\text{train}}]} \left[ (y_i - \mathbf{a}^\top \sigma(W_t \mathbf{x}_i)) \mathbf{a} \odot \sigma'(W_t \mathbf{x}_i) \right] \mathbf{x}_i^\top$$

and $\sigma'(\cdot)$ denotes the first order derivative of $\sigma(\cdot)$.

**Assumption C.1.** We adopt the following assumptions used in (Montanari et al., 2025; Ba et al., 2022).

1. (Proportional limit) $P_{\text{train}}, d, N \to \infty$ with $\psi_1 = N/d$, $\psi_2 = P_{\text{train}}/d$, and $0 < \psi_1, \psi_2 < \infty$.

2. (Gaussian initialization) $[W_0]_{kj} \sim \mathcal{N}(0, 1/N)$ for each $k \in [N]$ and $j \in [d]$.

3. (Gaussian readout) $a_k \sim \mathcal{N}(0, 1/N)$ for each $k \in [N]$.

4. (Normalized activation) The non-linear activation function $\sigma(\cdot)$ has $O(1)$-bounded first three derivatives almost surely. In addition, $\mathbb{E}[\sigma(G)] = 0$ and $\mathbb{E}[G\sigma(G)] \neq 0$ for $G \sim \mathcal{N}(0, 1)$.

5. (Non-degenerate label function) Let $F : \mathbb{R} \to [0, 1]$ be a continuous function satisfying

$$\inf \{x : \Pr[T < x] > 0\} = -\infty \text{ and } \sup \{x : \Pr[T > x] > 0\} = \infty$$

where $T = YG$, $G \sim \mathcal{N}(0, 1)$, and $\Pr[Y = 1 \,|\, G] = 1 - \Pr[Y = -1 \,|\, G] = F(G)$.

*Setting* C.2. We consider the following data generation process. Let $F : \mathbb{R} \to [0, 1]$ be a function satisfying Assumption C.1. Let $\beta_* \in \mathbb{R}^d$ be a hidden vector with $\|\beta_*\|_2 = 1$. The data distribution $\mathcal{D}_F(\beta_*)$ is defined by the following two steps: (i) sample $\mathbf{x} \sim \mathcal{N}(0, I_d)$, and (ii) sample $y$ with $\Pr[y = 1] = 1 - \Pr[y = -1] = F(\langle \beta_*, \mathbf{x} \rangle)$. Finally, the prediction accuracy of a network is defined as the expected accuracy of a fresh sample, i.e., $\Pr_{(\mathbf{x}, y) \sim \mathcal{D}_F(\beta_*)}[y f(\mathbf{x}) \geq 0]$.

**Parameter C.3.** *Given $\psi_1, \psi_2, F, \beta_*$ from Assumption C.1 and Setting C.2. We define the following parameters.*

$$\gamma_1 = \mathop{\mathbb{E}}_{G \sim \mathcal{N}(0,1)} [G\sigma(G)]$$

$$\gamma_2^2 = \mathop{\mathbb{E}}_{G \sim \mathcal{N}(0,1)} [\sigma(G)^2] - \mathop{\mathbb{E}}_{G \sim \mathcal{N}(0,1)} [G\sigma(G)]^2$$

$$\theta_1 = \mathop{\mathbb{E}}_{X \sim \mu_{\psi_1}} \left[ \frac{\gamma_1^2}{\gamma_1^2 X + \gamma_2^2} \right]$$

$$\theta_2 = \psi_1 \mathop{\mathbb{E}}_{X \sim \mu_{\psi_1}} \left[ \frac{\gamma_1^2 X}{\gamma_1^2 X + \gamma_2^2} \right]$$

$$\theta_3 = \mathop{\mathbb{E}}_{(G,Y) \sim \mathcal{D}_F} [YG]$$

$$\theta_4 = \left( \frac{1}{\psi_2} + \mathop{\mathbb{E}}_{(G,Y),(G',Y') \overset{i.i.d.}{\sim} \mathcal{D}_F} [YY'GG'] \right)$$

*where $\mu_{\psi_1}$ is the Marchenko-Pastur distribution with the ratio parameter being $\psi_1$ and $(G,Y) \sim \mathcal{D}_F$ is defined as the sampling process: $G \sim \mathcal{N}(0,1)$ and $\Pr[Y = 1] = 1 - \Pr[Y = -1] = F(G)$.*

**Theorem C.4.** *Given Assumption C.1 and consider $0 < \psi_1, \psi_2, \eta < \infty$.*

1. *(Capacity tracks the degree of feature learning) The storage capacity of 2-layer network trained with synthetic data defined in Setting C.2 after one gradient step is $\alpha_{P_{train}, d, N}(\psi_1, \psi_2, \eta)$ and*

$$\alpha_{P_{train}, d, N}(\psi_1, \psi_2, \eta) \xrightarrow{P_{train}, d, N \to \infty} \alpha(\psi_1, \psi_2, \eta)$$

   *Here the function $\alpha(\cdot)$ is defined as*

$$\alpha(\psi_1, \psi_2, \eta) = \left( \min_{c \in \mathbb{R}} \mathop{\mathbb{E}}_{(Z,G,Y) \sim \mathcal{D}_{\psi_1, \psi_2, \eta}} \left[ (-cYG - Z)_+^2 \right] \right)^{-1}$$

   *where $(Z, G, Y) \sim \mathcal{D}_{\psi_1, \psi_2, \eta}$ is defined as the following sampling process*

$$Z \sim \mathcal{N}(0,1), \; G \sim \mathcal{N}(0,1), \; \Pr[Y=1] = 1 - \Pr[Y=-1] = f_{\tau(\psi_1, \psi_2, \eta)}(G)$$

   *and the scalar function $f_\tau(\cdot)$ and $\tau(\psi_1, \psi_2, \eta)$ are defined as*

$$f_\tau(G) = \mathop{\mathbb{E}}_{G' \sim \mathcal{N}(0,1)} \left[ F(\sqrt{1 - \tau^2} G + \tau G') \right]$$

   *and*

$$\tau = \tau(\psi_1, \psi_2, \eta) = \sqrt{\tau_0(\psi_1, \psi_2)^2 - \tau_\Delta(\psi_1, \psi_2, \eta)^2}$$

   *where $\tau_0(\cdot)$ and $\tau_\Delta(\cdot)$ are scalar functions defined as*

$$\tau_0(\psi_1, \psi_2)^2 = 1 - \theta_2$$

   *and*

$$\tau_\Delta(\psi_1, \psi_2, \eta)^2 = \frac{\eta^2 \theta_1 (1 - \theta_2)^2 \theta_3^2}{1 + \eta^2 \theta_1 (1 - \theta_2) \theta_4}$$

   *where the parameters $\theta_i$'s are defined in Parameter C.3. In particular, $0 < \alpha(\psi_1, \psi_2, \eta) < \alpha(\psi_1, \psi_2, \eta')$ for all $0 < \eta < \eta'$.*

2. *(Capacity analytically links to prediction accuracy) The prediction accuracy of 2-layer network trained with synthetic data defined in Setting C.2 after one gradient step is $\text{Acc}_{P_{train}, d, N}(\psi_1, \psi_2, \eta)$ and*

$$\text{Acc}_{P_{train}, d, N}(\psi_1, \psi_2, \eta) \xrightarrow{P_{train}, d, N \to \infty} \text{Acc}(\psi_1, \psi_2, \eta)$$

*Here the function $\mathsf{Acc}(\cdot)$ is defined as*

$$\mathsf{Acc}(\psi_1, \psi_2, \eta) = \mathop{\mathbb{E}}_{(G,Y) \sim \mathcal{D}_F} \left[ \Phi \left( \frac{\eta \gamma_1^2 \theta_3}{\sqrt{\frac{\eta^2 \gamma_1^4}{\psi_2} + \gamma_1^2 + \gamma_*^2}} YG \right) \right]$$

*In particular, there exists an increasing and invertible function $g_{\psi_1, \psi_2} : [0, 1] \to \mathbb{R}_+$ such that*

$$\mathsf{Acc}(\psi_1, \psi_2, \eta) = g_{\psi_1, \psi_2}(\alpha(\psi_1, \psi_2, \eta)).$$

## C.2. Proof for Theorem Theorem C.4

**Step 1: Rank-1 approximation of gradient descent in 2-layer networks by ref. (Ba et al., 2022).** When the learning rate is constant, i.e., $\eta = O(1)$, ref. (Ba et al., 2022) shows that the gradient update matrix can be approximated by a rank-1 matrix. In particular, the following is a restatement of Proposition 2 in (Ba et al., 2022).

**Proposition C.5** (Proposition 2 in (Ba et al., 2022))**.** *Given Assumption C.1 and Setting C.2, there exist some constants $c, C > 0$ such that for all large $P_{train}, N, d$, the following holds*

$$\left\| G_0 - \gamma_1 \mathbf{a} \left( \frac{\sum_i y_i \mathbf{x}_i^\top}{P_{train}} \right) \right\| \leq \frac{C \log^2 P_{train}}{\sqrt{P_{train}}} \cdot \|G_0\|$$

*with probability at least $1 - P_{train} e^{-c \log^2 P_{train}}$ and $\| \cdot \|$ denotes the operator norm.*

**Step 2: A formula for the storage capacity of a Gaussian model by ref. (Montanari et al., 2025).** The storage capacity of a Gaussian model is proven in (Montanari et al., 2025). In particular, the following is a restatement of the Proposition 5.1 in (Montanari et al., 2025).

**Definition C.6** (Gaussian model)**.** Let $\theta_* \in \mathbb{R}^N$ be some latent vector. A sample $(\mathbf{x}_i, y_i) \in \mathbb{R}^N \times \{\pm 1\}$ is i.i.d. sampled as follows. First, sample $\mathbf{x}_i$ from $\mathcal{N}(0, \Sigma)$ where $\Sigma$ is a covariance matrix satisfying certain technical condition as defined in Assumption 1-2 in (Montanari et al., 2025). Next, let $y_i = +1$ with probability $f(\langle \theta_*, \mathbf{x}_i \rangle)$ for some function $f$ satisfying Assumption 3 in (Montanari et al., 2025).

**Proposition C.7** (Theorem 3 in (Montanari et al., 2025))**.** *Consider a Gaussian model satisfying Definition C.6. As $P_{train}, N, d \to \infty$, the storage capacity converges to*

$$\alpha^* = \left( \min_{c \in \mathbb{R}} \mathop{\mathbb{E}}_{(Z,G,Y) \sim \mathcal{D}_f} \left[ (-cYG - Z)_+^2 \right] \right)^{-1}$$

*where $(Z, G, Y) \sim \mathcal{D}_f$ is defined as the following sampling process*

$$Z \sim \mathcal{N}(0, 1), \ G \sim \mathcal{N}(0, 1), \ \Pr[Y = 1] = 1 - \Pr[Y = -1] = f(\rho \cdot G).$$

*where $\rho$ is some scalar related to the Gaussian model as defined in Assumption 2 of (Montanari et al., 2025).*

Note that the capacity only depends on the alignment between data and task (as encoded in $f$) and does not depend on the covariance structure. The dependence on the covariance structure will appear when one considers the non-zero margin version of capacity.

**Step 3: A Gaussian equivalent model for 2-layer NNs after one gradient step.** Next, we combine a Gaussian equivalent model for random feature 2-layer NNs in (Montanari et al., 2025) (Theorem 3) and the rank-1 approximation of gradient step in Proposition C.5 to get a Gaussian equivalent model for 2-layer NNs after one gradient step.

**Proposition C.8.** *Given Assumption C.1 and $0 < \psi_1, \psi_2, \eta < \infty$. Let $d \in \mathbb{N}$ and $(W_1, \beta_*, F)$ be the weight matrix, hidden*

*vector, and label function from Setting C.2. Let* $\alpha_{P_{train},d,N}^{GM}(\psi_1, \psi_2, \eta)$ *be the capacity of the following Gaussian model:*

$$
\begin{aligned}
\Sigma_{d,\eta} &= \gamma_1^2 W_1 W_1^\top + \gamma_*^2 I \\
\theta_{*,d,\eta} &= \alpha_{d,\eta}^{-1}\gamma_1(\gamma_1^2 W_1 W_1^\top + \gamma_*^2 I)^{-1} W_1 \beta_* \\
\alpha_{d,\eta}^2 &= \gamma_1^2 \beta_*^\top W_1^\top (\gamma_1^2 W_1 W_1^\top + \gamma_*^2 I)^{-1} W_1 \beta_* \\
\tau_{d,\eta}^2 &= 1 - \alpha_{d,\eta}^2 \\
f_{d,\eta}(x) &= \mathop{\mathbb{E}}_{G\sim\mathcal{N}(0,1)}\left[F(\alpha_{d,\eta}x + \tau_{d,\eta}G)\right].
\end{aligned}
\tag{7}
$$

*We have that*

$$
\lim_{P_{train},d,N\to\infty} |\alpha_{P_{train},d,N}(\psi_1,\psi_2,\eta) - \alpha_{P_{train},d,N}^{GM}(\psi_1,\psi_2,\eta)| = 0
$$

*and*

$$
\alpha_{P_{train},d,N}^{GM}(\psi_1,\psi_2,\eta) \xrightarrow{P_{train},d,N\to\infty} \alpha(\psi_1,\psi_2,\eta).
$$

*Here the function* $\alpha(\cdot)$ *is defined as*

$$
\alpha(\psi_1,\psi_2,\eta) = \left(\min_{c\in\mathbb{R}} \mathop{\mathbb{E}}_{(Z,G,Y)\sim\mathcal{D}_{f_\tau(\psi_1,\psi_2,\eta)}} \left[(-cYG - Z)_+^2\right]\right)^{-1}
$$

*where the scalar function* $f_\tau(\cdot)$ *and* $\tau(\psi_1,\psi_2,\eta)$ *are defined as*

$$
f_\tau(G) = \mathop{\mathbb{E}}_{G'\sim\mathcal{N}(0,1)} \left[F(\sqrt{1-\tau^2}G + \tau G')\right]
$$

*and*

$$
\tau = \tau(\psi_1,\psi_2,\eta) = \lim_{d\to\infty} \tau_{d,\eta} = \sqrt{\tau_0(\psi_1,\psi_2)^2 - \tau_\Delta(\psi_1,\psi_2,\eta)^2}.
$$

*where* $\tau_0(\psi_1,\psi_2) = \lim_{d\to\infty} \tau_{d,0}$.

To derive the Gaussian equivalent model in Proposition C.8 of the random features model after one gradient step defined in Setting C.2, we analyze the following random features and their associated labels:

$$
\Phi_0(\mathbf{x}_i) = \sigma(W_1\mathbf{x}_i), \quad \Pr[y_i = 1|\mathbf{x}_i] = 1 - \Pr[y_i = -1|\mathbf{x}_i] = F(\langle\beta_*,\mathbf{x}_i\rangle), \quad \|\beta_*\|_2 = 1
$$

where $\mathbf{x}_i \sim \mathcal{N}(0, I_d)$ and $W_1 = W_0 + \eta G_0$ while $G_0$ satisfies the bound given in Proposition C.5. Given the assumptions in Assumption C.1, we can decompose the nonlinear activation function $\sigma$ into Hermite polynomials. Following our parameters in Parameter C.3, we define the Gaussian equivalent features of our model as the linearization of Equation C.2:

$$
\mathbf{g}_i = \gamma_1 W_1 \mathbf{x}_i + \gamma_2 \mathbf{h}_i
$$

where $\mathbf{h}_i \sim \mathcal{N}(0, I_N)$ are independent from everything else. Now, we wish to find a similar linearized Gaussian model for the labels $y_i$ given the Gaussian equivalent features $\mathbf{g}_i$. It is easy to check that the Gaussian features has the following covariance:

$$
\mathbf{g}_i \sim \mathcal{N}(0, \Sigma_{d,\eta}), \quad \Sigma_{d,\eta} = \gamma_1^2 W_1 W_1^\top + \gamma_*^2 I
$$

By matching covariance through Equation C.2, we obtain

$$
\mathbf{x}_i = \gamma_1 W_1^\top \Sigma_{d,\eta}^{-1} \mathbf{g}_i + Q^{1/2}\tilde{\mathbf{h}}_i
$$

where $Q = \gamma_2^2(\gamma_2^2 I_N + \gamma_1^2 W_1^\top W_1)^{-1}$ and $\tilde{\mathbf{h}}_i \sim \mathcal{N}(0, I_N)$ are independent of $\mathbf{x}_i$. Therefore, we can rewrite the label function parameter as

$$
\langle\beta_*,\mathbf{x}_i\rangle = \alpha_{d,\eta}\langle\theta_{*,d,\eta},\mathbf{g}_i\rangle + \varepsilon_i
$$

where $\varepsilon_i \sim \mathcal{N}(0, \tau_{d,\eta}^2)$ are independent of $\mathbf{g}_i$. Effectively, we obtain an equivalent label function

$$
f_{d,\eta}(x) = \mathop{\mathbb{E}}_{G\sim\mathcal{N}(0,1)}[F(\alpha_{d,\eta}x + \tau_{d,\eta}G)]
$$

such that $\Pr[y_i = 1|\mathbf{x}_i] = 1 - \Pr[y_i = -1|\mathbf{x}_i] = f_{d,\eta}(\langle\theta_{*,d,\eta},\mathbf{g}_i\rangle)$. It is easy to verify that this Gaussian model satisfies the assumptions in Definition C.6.

**Step 4: Analysis of $\tau$.** Finally, we combine Proposition C.5 and Proposition C.8 to get the formula for the right hand side of Equation 7. From Proposition C.5, we approximate $W_1$ as $W_1 = W_0 + \mathbf{a}\mathbf{u}^\top$ where $\mathbf{u} = \eta \sum_i y_i \mathbf{x}_i^\top / P_{\text{train}}$. To rewrite the right hand side of Equation 7, we first deal with the matrix inverse term using the same trick as in ref. (Ba et al., 2022). Let $\Sigma_t = \gamma_1^2 W_t W_t^\top + \gamma_*^2 I$. Observe that

$$\Sigma_1 = \Sigma_0 + \gamma_1^2 \begin{bmatrix} \mathbf{a} & \mathbf{c} \end{bmatrix} \begin{bmatrix} L_1 & 1 \\ 1 & 0 \end{bmatrix} \begin{bmatrix} \mathbf{a}^\top \\ \mathbf{c}^\top \end{bmatrix}$$

where $\mathbf{c} = W_0 \mathbf{u}$. By Sherman-Morrison-Woodbury formula, we have

$$\Sigma_1^{-1} = \Sigma_0^{-1} - \gamma_1^2 \Sigma_0^{-1} \begin{bmatrix} \mathbf{a} & \mathbf{c} \end{bmatrix} \left( \begin{bmatrix} L_1 & 1 \\ 1 & 0 \end{bmatrix}^{-1} + \gamma_1^2 \begin{bmatrix} \mathbf{a}^\top \\ \mathbf{c}^\top \end{bmatrix} \Sigma_0^{-1} \begin{bmatrix} \mathbf{a} & \mathbf{c} \end{bmatrix} \right)^{-1} \begin{bmatrix} \mathbf{a}^\top \\ \mathbf{c}^\top \end{bmatrix} \Sigma_0^{-1}$$

$$= \Sigma_0^{-1} - \Delta_{aa} - \Delta_{cc} + \Delta_{ac} + \Delta_{ca}$$

where

$$\Delta_{aa} = \gamma_1^2 \frac{L_4 - L_1}{D} \Sigma_0^{-1} \mathbf{a}\mathbf{a}^\top \Sigma_0^{-1}$$

$$\Delta_{cc} = \gamma_1^2 \frac{L_3}{D} \Sigma_0^{-1} \mathbf{c}\mathbf{c}^\top \Sigma_0^{-1}$$

$$\Delta_{ac} = \gamma_1^2 \frac{1 + L_6}{D} \Sigma_0^{-1} \mathbf{a}\mathbf{c}^\top \Sigma_0^{-1}$$

$$\Delta_{ca} = \gamma_1^2 \frac{1 + L_6}{D} \Sigma_0^{-1} \mathbf{c}\mathbf{a}^\top \Sigma_0^{-1}$$

and

$$L_0 = \gamma_1^2 \beta_*^\top W_0^\top \Sigma_0^{-1} W_0 \beta_*$$
$$L_1 = \mathbf{u}^\top \mathbf{u}$$
$$L_2 = \mathbf{u}^\top \beta_*$$
$$L_3 = \gamma_1^2 \mathbf{a}^\top \Sigma_0^{-1} \mathbf{a}$$
$$L_4 = \gamma_1^2 \mathbf{c}^\top \Sigma_0^{-1} \mathbf{c}$$
$$L_5 = \gamma_1^2 \mathbf{c}^\top \Sigma_0^{-1} W_0 \beta_*$$
$$L_6 = \gamma_1^2 \mathbf{a}^\top \Sigma_0^{-1} \mathbf{c}$$
$$L_7 = \mathbf{a}^\top \mathbf{c}$$
$$L_8 = \gamma_1^2 \mathbf{a}^\top \Sigma_0^{-1} W_0 \beta_*$$
$$D = L_3(L_4 - L_1) - (1 + L_6)^2$$

Thus, we can rewrite the right hand side of Equation 7 as follows.

$$\begin{aligned}
\tau_{d,\eta} = {} & 1 - \gamma_1^2 \beta_*^\top (W_0 + \mathbf{a}\mathbf{u}^\top)^\top \Sigma_0^{-1} (W_0 + \mathbf{a}\mathbf{u}^\top) \beta_* \\
& + \gamma_1^2 \beta_*^\top (W_0 + \mathbf{a}\mathbf{u}^\top)^\top \Delta_{aa} (W_0 + \mathbf{a}\mathbf{u}^\top) \beta_* \\
& + \gamma_1^2 \beta_*^\top (W_0 + \mathbf{a}\mathbf{u}^\top)^\top \Delta_{cc} (W_0 + \mathbf{a}\mathbf{u}^\top) \beta_* \\
& - \gamma_1^2 \beta_*^\top (W_0 + \mathbf{a}\mathbf{u}^\top)^\top \Delta_{ac} (W_0 + \mathbf{a}\mathbf{u}^\top) \beta_* \\
& - \gamma_1^2 \beta_*^\top (W_0 + \mathbf{a}\mathbf{u}^\top)^\top \Delta_{ca} (W_0 + \mathbf{a}\mathbf{u}^\top) \beta_* \\
= {} & 1 - L_0 - L_2^2 L_3 - 2 L_2 L_8 \\
& + \frac{L_4 - L_1}{D} (L_2 L_3 + L_8)^2 \\
& + \frac{L_3}{D} (L_5 + L_2 L_6)^2 \\
& - 2 \frac{1 + L_6}{D} (L_2 L_3 + L_8)(L_5 + L_2 L_6).
\end{aligned}$$

Similar to Proposition 29 in (Ba et al., 2022), by Hanson-Wright inequality, we have that $L_6, L_8, L_7 \to 0$.

$$L_0 \to \theta_2$$
$$L_1 \to \eta^2 \theta_4$$
$$L_2 = \eta \theta_3$$
$$L_3 \to \gamma_1^2 \mathop{\mathbb{E}}_{X \sim \mu_{\psi_1}} \left[ \frac{1}{\gamma_1^2 X + \gamma_2^2} \right] = \theta_1$$
$$L_4 \to \gamma_1^2 \eta^2 \theta_4 \cdot \psi_1 \mathop{\mathbb{E}}_{X \sim \mu_{\psi_1}} \left[ \frac{X}{\gamma_1^2 X + \gamma_2^2} \right] = \eta^2 \theta_2 \theta_4$$
$$L_5 \to \gamma_1^2 \eta \theta_3 \cdot \psi_1 \mathop{\mathbb{E}}_{X \sim \mu_{\psi_1}} \left[ \frac{X}{\gamma_1^2 X + \gamma_2^2} \right] = \eta \theta_2 \theta_3$$
$$L_6, L_7, L_8 \to 0$$
$$D \to L_3(L_4 - L_1) - 1 \to \eta^2 \theta_1 (\theta_2 - 1)\theta_4 - 1$$

To sum up, we have

$$\begin{aligned}
\lim_{d \to \infty} \tau_{d,\eta} = {} & 1 - \theta_2 - \frac{\eta^2 \theta_1 \theta_3^2 (\eta^2 \theta_1 (\theta_2 - 1)\theta_4 - 1)}{\eta^2 \theta_1 (\theta_2 - 1)\theta_4 - 1} \\
& + \frac{\eta^4 \theta_1^2 (\theta_2 - 1)\theta_3^2 \theta_4}{\eta^2 \theta_1 (\theta_2 - 1)\theta_4 - 1} \\
& + \frac{\theta_1 \theta_2^2 \theta_3^2}{\eta^2 \theta_1 (\theta_2 - 1)\theta_4 - 1} \\
& - 2 \frac{\eta^2 \theta_1 \theta_2 \theta_3^2}{\eta^2 \theta_1 (\theta_2 - 1)\theta_4 - 1} \\
= {} & 1 - \theta_2 - \frac{\eta^2 \theta_1 (1 - \theta_2)^2 \theta_3^2}{1 + \eta^2 \theta_1 (1 - \theta_2)\theta_4}.
\end{aligned}$$

This completes the proof for the first part of Theorem C.4.

**Step 5: Analysis for prediction accuracy.** Recall from Setting C.2 the definition of prediction accuracy of the network after a gradient step is $\Pr_{(\mathbf{x},y) \sim \mathcal{D}_F(\beta_*)}[y \mathbf{a}^\top \sigma(W_1 \mathbf{x}) \geq 0]$. By Gaussian equivalence and Proposition C.5, we have that the following.

$$\begin{aligned}
& \mathsf{Acc}_{P_{\text{train}}, d, N}(\psi_1, \psi_2, \eta) \\
& = \Pr_{\substack{(\mathbf{x},y) \sim \mathcal{D}_F(\beta_*) \\ \mathbf{a}, W_1}} [y \mathbf{a}^\top \sigma(W_1 \mathbf{x}) \geq 0].
\end{aligned}$$

By Proposition C.5, we can further approximate the equation as follows.

$$= \Pr_{\substack{(\mathbf{x},y) \sim \mathcal{D}_F(\beta_*) \\ \mathbf{a}, W_0, \mathbf{u}}} [y \mathbf{a}^\top \sigma((W_0 + \mathbf{a}\mathbf{u}^\top)\mathbf{x}) \geq 0] + o(1).$$

By Gaussian equivalence, we can further approximate the equation as follows.

$$= \Pr_{\substack{(\mathbf{x},y) \sim \mathcal{D}_F(\beta_*) \\ \mathbf{a}, W_0, W_*, \mathbf{u}}} [y \mathbf{a}^\top (\gamma_1 (W_0 + \mathbf{a}\mathbf{u}^\top) + \gamma_* W_*)\mathbf{x}) \geq 0] + o(1)$$

where $W_* \in \mathbb{R}^{N \times d}$ and $([W_*]_{kj} \sim \mathcal{N}(0, 1/N))$ for each $k \in [N], j \in [d]$. Note that as $\mathbf{a}, W_0, W_*$ are independent, we can further simplify the equation as follows.

$$= \Pr_{\substack{(\mathbf{x},y) \sim \mathcal{D}_F(\beta_*) \\ \mathbf{a}, W_*', \mathbf{u}}} [y\gamma_1 \mathbf{u}^\top \mathbf{x} + \sqrt{\gamma_1^2 + \gamma_*^2} \cdot y \mathbf{a}^\top W_*' \mathbf{x} + o(1) \geq 0] + o(1)$$

where $W'_* \in \mathbb{R}^{N \times d}$ and $([W'_*]_{kj} \sim \mathcal{N}(0, 1/N))$ for each $k \in [N], j \in [d]$. Note that as $\mathbf{a}, W'_*$ are independent, we can further simplify the equation as follows.

$$= \Pr_{\substack{(\mathbf{x},y) \sim \mathcal{D}_F(\beta_*) \\ Z \sim \mathcal{N}(0,1)}} \left[ \eta\gamma_1^2 \mathbb{E}_{(\mathbf{x}',y') \sim \mathcal{D}_F(\beta_*)} [yy'\mathbf{x}'^\top \mathbf{x}] + \sqrt{\gamma_1^2 + \gamma_*^2} \cdot Z + o(1) \geq 0 \right] + o(1).$$

Note that by decomposing $\mathbf{x}$ and $\mathbf{x}'$ to direction that's parallel to $\beta_*$ and orthogonal to $\beta_*$, we can further simplify the equation as follows.

$$= \Pr_{\substack{(G,Y) \sim \mathcal{D}_F \\ Z,Z' \sim \mathcal{N}(0,1)}} \left[ \eta\gamma_1^2 \left( \mathbb{E}_{(G',Y') \sim \mathcal{D}_F} [YY'GG'] + \sqrt{1/\psi_2} Z' \right) + \sqrt{\gamma_1^2 + \gamma_*^2} \cdot Z + o(1) \geq 0 \right] + o(1)$$

$$= \Pr_{\substack{(G,Y) \sim \mathcal{D}_F \\ Z \sim \mathcal{N}(0,1)}} \left[ \eta\gamma_1^2 \theta_3 YG + \sqrt{\frac{\eta^2\gamma_1^4}{\psi_2} + \gamma_1^2 + \gamma_*^2} \cdot Z + o(1) \geq 0 \right] + o(1)$$

$$= \mathbb{E}_{(G,Y) \sim \mathcal{D}_F} \left[ \Phi \left( \frac{\eta\gamma_1^2 \theta_3}{\sqrt{\frac{\eta^2\gamma_1^4}{\psi_2} + \gamma_1^2 + \gamma_*^2}} YG \right) \right] + o(1).$$

Note that when fixing $\psi_1, \psi_2$ and non-trivial $F$, both capacity formula and prediction accuracy formula are increasing and invertible with respect to $\eta$. As a consequence, the two quantities are also analytically connected by an increasing and invertible function. This completes the proof for the second part of Theorem C.4. We also provide numeric checks for the formulas in Figure 13.

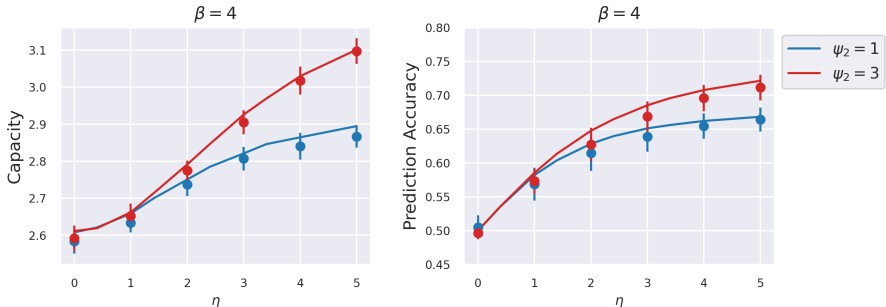

*Figure 13.* Numerical checks for the formulas in Theorem C.4. We run the simulation with $d = 2000, \psi_1 = 1$, ReLU activation, and label function $f(x) = \frac{1}{1+e^{-4x}}$ for 50 repetitions. Left: numerical checks for the capacity formula. Right: numerical checks for the prediction accuracy formula.

# D. 2-Layer Non-linear Neural Networks

In this paper, we use 2-layer non-linear neural networks and Gaussian mixture models (for input data generation) as a convenient experimental setup to systematically explore different regimes in feature learning. Moreover, given its medium level of complexity, it might be possible to have an analytical characterization of our numerical findings, and we leave it as an interesting future direction.

## D.1. Experimental setup

### D.1.1. SYNTHETIC DATA GENERATION

We focus on point manifold, which consists of data points associated with the same label. As discussed in the previous section, we are particularly interested in the effective radius, dimension, center alignment, axes alignment, and center-axes alignment of the representation manifolds. Therefore, we consider a synthetic model to generate training and test data with relevant geometric interpretations. Namely, construct $P \in \mathbb{N}$ synthetic data manifolds with radius $R \in \mathbb{R}_+$, intrinsic

dimension $D \in \mathbb{N}$, size $M \in \mathbb{N}$. The manifold layouts are further determined by center correlation strength $\rho_C \in [0, 1)$, axes correlation strength $\rho_A \in [0, 1)$, and center-axes correlation strength $\psi \in [0, 1)$, all of which we would detail in the following subsections.

**Isotropic spherical manifolds.** First, we consider the simplest case: manifolds with isotropic Gaussian center distribution and axes distribution with no correlations. This is the scenario considered in Section 3 and Section 4.

Let $d \in \mathbb{N}$ be the dimension of the data. We consider $P$ point manifolds $\{\mathcal{M}_i\}_{i \in [P]}$ with manifold size $M \in \mathbb{N}$ and radius $R$ that lies in a subspace of dimension $D$. Each manifold is defined as

$$\mathcal{M}_i = \{\mathbf{u}_0 + R \cdot \sum_{j=1}^{D} s_j^k \mathbf{u}_j + \epsilon \mathbf{v}_k\}_{k \in [M]}$$

where the axes $\mathbf{u}_j \sim N(0, I_d/d)$, the coordinates $s_j^k \sim N(0, 1)$, the noise vectors $\mathbf{v}_k \sim N(0, I_d/d)$, and $\epsilon = 10^{-2}$. The pre-scaled points in the manifolds $\{\sum_{j=1}^{D} s_j^k u_j\}_{k \in [M]}$ are well-normalized to unit norm.

Test manifolds share the same model except that the noise vectors $\mathbf{v}_j$ are sampled again in the same distribution.

**Isotropic Gaussian manifolds.** In certain experiments, we drop in the intrinsic dimension $D$ and directly consider manifolds defined as

$$\mathcal{M}_i = \{\mathbf{u}_0 + R \cdot \mathbf{v}_k\}_{k \in [M]}$$

where the noise vectors are $\mathbf{v}_k \sim N(0, I_d/d)$. Test manifolds share the same model except that the noise vectors $\mathbf{v}_k$ are sampled again in the same distribution.

**Correlated spherical manifolds.** To generated correlated manifolds, we consider an auto-regressive model described by the covariance matrix $C = (\rho^{|i-j|})_{ij} \in \mathbb{R}^{P \times P}$, where $\rho \in [0, 1)$ is either the center correlation strength $\rho_C$ or axes correlation strength $\rho_A$. The center covariance $C_C$ is then mixed into the isotropic manifold centers $\{\mathbf{u}_0^j \sim N(0, I_d/d)\}_{j \in [M]}$. The axes covariance matrices $C_A^i$ is mixed into the isotropic axes $\{\mathbf{u}_i^j \sim N(0, I_d/d)\}_{j \in [M]}$ for each $i = 1, 2, \ldots, D$ respectively. The mixing is performed through multiplying the column matrix $M_C$ or $M_A^j \in \mathbb{R}^{P \times d}$ of centers or each axes with the Cholesky decomposition of $C_C$ or $C_A^i$. To incorporate center-axes correlation, we scale each center vector $\mathbf{u}_0$ by a factor of $(1 + \psi \cdot q)$ where $q \sim N(0, 1)$.

**Labels.** For $P$ manifolds with manifold size $M$, the $P$ labels are randomly sampled from a uniform distribution on $\{\pm 1\}$. Each label is associated with $M$ data points in the individual manifold. When learning with binary cross entropy, the labels are reassigned as $\{0, 1\}$ during loss and gradient computation.

D.1.2. 2-LAYER NEURAL NETWORK ARCHITECTURE

The model architecture we consider is similar to the architecture mentioned in Appendix C.

Let $d \in \mathbb{N}$ be the input data dimension, $N \in \mathbb{N}$ be the number of hidden neurons, $K \in \mathbb{N}$ be the number of linear readouts, $\alpha \in \mathbb{R}_+$ be the scaling factor of the readout weights.

Let $W = W_0 \in \mathbb{R}^{N \times d}$ be the initial weight matrix of a fully connected 2-layer neural network. Let $\{a_0^i\}_{i \in [K]}$ be a list of initial readout weights where $a_0^i \in \mathbb{R}^N$. Let $\sigma(\cdot) : \mathbb{R} \to \mathbb{R}$ be a non-linear activation function, e.g. ReLU or $\tanh$.

The feature of an input vector is defined as $\phi(\mathbf{x}) = \sigma(W\mathbf{x})$. The 2-layer neural network parameterized by $W$ and $a^i$ is defined as

$$f(W, a^i; \mathbf{x}) = \frac{\alpha}{\sqrt{N}} \mathbf{a}^\top \phi(\mathbf{x})$$

where the label prediction for data point $\mathbf{x}$ is $\mathsf{sgn}(f(\mathbf{x}))$ when learning with the mean squared error loss function. When learning with binary cross entropy loss function, we use $\{0, 1\}$ as labels and $\varsigma(f(\mathbf{x}))$ as prediction instead, where $\varsigma$ is the standard sigmoid function.

### D.1.3. LEARNING RULE

**Loss function and gradient update.** Let $\eta \in \mathbb{R}_+$ be the learning rate of the weight matrix, $c \in \mathbb{R}_+$ be the scaling factor of the readout learning rate, and let $\{(\mathbf{x}_i, y_i)\}_{i \in [PM]}$ be the collection of training data, where $P$ is the number of manifolds and $M$ is the manifold size.

We consider gradient descent over the loss function

$$\mathcal{L}(f) = \frac{1}{\alpha^2} \frac{1}{PM} \sum_{i \in [PM]} \ell(f(\mathbf{x}_i), y_i)$$

where $\ell : \mathbb{R} \times \{\pm 1\} \to \mathbb{R}$ is either the mean squared error (MSE)

$$\ell_{MSE}(z, y) = \frac{1}{2}(z - y)^2$$

or $l : \mathbb{R} \times \{0, 1\} \to \mathbb{R}$ is the binary cross entropy (BCE)

$$\ell_{BCE}(z, y) = y \cdot \log(1 + e^{-z}) + (1 - y) \cdot \log(1 + e^z)$$

**Mean squared error.** For the weight matrix, the gradient update with learning rate $\eta > 0$ is $W_{t+1} = W_t + \eta G_t$ where

$$G_t = \frac{1}{\alpha^2} \frac{1}{PM} \sum_{i \in [PM]} \frac{1}{K} \sum_{j \in [K]} \left[ (y_i - \frac{\alpha}{\sqrt{N}} \mathbf{a}_t^{j\top} \sigma(W_t \mathbf{x}_i)) \frac{\alpha}{\sqrt{N}} \mathbf{a}_t^j \odot \sigma'(W_t \mathbf{x}_i) \right] \mathbf{x}_i^\top$$

and $\sigma'(\cdot)$ denotes the first order derivative of $\sigma(\cdot)$. For each linear readout, the gradient update is $a_{t+1} = a_t + c\eta g_t$ where

$$g_t = \frac{1}{\alpha^2} \frac{1}{PM} \sum_{i \in [PM]} \left[ y_i - \frac{\alpha}{\sqrt{N}} \mathbf{a}_t^\top \sigma(W_t \mathbf{x}_i) \right] \frac{\alpha}{\sqrt{N}} \sigma(W_t \mathbf{x}_i)$$

Note that the $\alpha^{-2}$ multiplier on the loss function to ensure common convergence time when $\alpha \to \infty$ as mentioned in (Geiger et al., 2020).

**Binary cross entropy.** For the weight matrix, the gradient update with learning rate $\eta > 0$ is $W_{t+1} = W_t + \eta G_t$ where

$$G_t = \frac{1}{\alpha^2} \frac{1}{PM} \sum_{i \in [PM]} \frac{1}{K} \sum_{j \in [K]} \left[ (y_i - \varsigma[\frac{\alpha}{\sqrt{N}} \mathbf{a}_t^{j\top} \sigma(W_t \mathbf{x}_i)]) \frac{\alpha}{\sqrt{N}} \mathbf{a}_t^j \odot \sigma'(W_t \mathbf{x}_i) \right] \mathbf{x}_i^\top$$

where $\varsigma$ denotes the standard sigmoid function and $\sigma$ denotes the activation function. For each linear readout, the gradient update is $a_{t+1} = a_t + c\eta g_t$ where

$$g_t = \frac{1}{\alpha^2} \frac{1}{PM} \sum_{i \in [PM]} \left[ y_i - \varsigma[\frac{\alpha}{\sqrt{N}} \mathbf{a}_t^\top \sigma(W_t \mathbf{x}_i)] \right] \frac{\alpha}{\sqrt{N}} \sigma(W_t \mathbf{x}_i)$$

If not otherwise noted, we conduct experiments with the MSE loss function and $\mathrm{ReLU}$ activation function by default.

**A Note on Learning rate.** We define $\bar{\eta} = \eta \alpha^{-1}$ as the normalized effective learning rate. During training, We implicitly scale the learning rate $\eta$ by a factor of $\sqrt{N}$ in the experiments to enter the rich regime as mentioned in (Ba et al., 2022).

### D.1.4. TRAINING

For each 2-layer neural network experiment conducted in the paper, forty random seeds are chosen from 0 to 39000 with an interval of 1000 to train forty models in parallel for $10^5$ epochs. All training are conducted on the Flatiron Institute high performance computing clusters.

### D.1.5. FEATURE EXTRACTION

During analysis, fifty epochs are sampled uniformly in log-scale. For each model at checkpoint epoch $t$, we extract total $P$ size $M$ manifold representations $\{\Phi_t(\mathbf{x}_i)\}_{i \in [PM]}$ associated with labels $\{y_i\}_{i \in [PM]}$. We perform conventional analysis and manifold capacity analysis described in Appendix A and Appendix B respectively. We will present more details in the following experiment sections.

### D.2. Capacity is a robust measure of feature learning across architecture, data, and learning rule variations

The purpose of this section is to support Section 3 by showcasing that capacity is able to quantify feature learning even when model architecture, data distribution, and learning rule varies.

### D.2.1. FEATURE ANALYSIS METHODS

Here, we briefly present the conventional feature analysis methods and capacity analysis method and how they are computed in the experimental setup.

**Representation level analysis.** Activation stability is a representation level metric that intuitively captures how much neurons are activated in hidden units. Formally, we define it as

$$\frac{\sum_{i=1}^{PM} \sum_{j=1}^{N} \mathbf{1}_{>0}(\phi_j(\mathbf{x}_i))}{PMN}$$

Another conventional method to disentangle feature learning at representation level is tracking the norm of deviation from initial weights (Jacot et al., 2018)

$$\frac{\|W_t - W_0\|}{\|W_0\|}$$

On the other hand, the cosine similarity (Liu et al., 2024) can be used to study alignment at representation level

$$\frac{\Phi_t \Phi_0}{\|\Phi_t\|\|\Phi_0\|}$$

where $(\Phi_t)_{ij} = \phi_t(\mathbf{x}_i) \cdot \phi_t(\mathbf{x}_j) \in \mathbb{R}^{PM \times PM}$ is the gram matrix of features over the test data.

**Kernel methods.** The kernel methods for quantifying feature learning involves computing the Neural Tangent Kernel (NTK) (Jacot et al., 2018) for each pair of test data points:

$$\Theta_t(\mathbf{x}_1, \mathbf{x}_2) = \nabla_{w_t} f(\mathbf{x}_1) \cdot \nabla_w f(\mathbf{x}_2)$$

where $\nabla_{w_t} f$ denotes the total gradient of the neural network at epoch $t$ with respect to the hidden weights $W_t$ and readout weights $\{a_t^j\}$. Note that we scale the readout contribution to the total gradient by the readout learning rate factor $c \in \mathbb{R}_+$ aforementioned. Hence,

$$\nabla_w f(\mathbf{x}) = \nabla_{W_t} f(\mathbf{x}) + \frac{1}{K} \sum_{j=1}^{K} \nabla_{a_t^j} f(\mathbf{x})$$

After obtaining the gram matrix $\Theta_t = \Theta_t(\mathbf{x}_i, \mathbf{x}_j)_{ij} \in \mathbb{R}^{PM \times PM}$ from the test data, we can compute the *NTK change* defined as

$$\frac{\|\Theta_t - \Theta_0\|}{\|\Theta_0\|}$$

which can be interpreted as the relative deviation of the the kernel from initialization in the Frobenius norm metric. Conventionally studied, NTK change disentangles lazy and feature learning, as detailed in (Jacot et al., 2018). We present NTK change in Section 3 Figure 4 to compare it with capacity as the metric to track feature learning.

The *kernel alignment* can be similarly defined as the cosine similarity of initial and current NTK gram matrices:

$$\frac{\Theta_t \Theta_0}{\|\Theta_t\|\|\Theta_0\|}$$

which can be interpreted as the relative deviation of the kernel from initialization in terms of alignment. Kernel alignment is also studied in (Liu et al., 2024) to disentangle lazy and feature learning.

The *centered kernel alignment* (Kornblith et al., 2019) is another approximation method to study kernel evolution when the gram matrices is large:

$$\frac{HSIC(\Theta_t, \Theta_0)}{\sqrt{HSIC(\Theta_t, \Theta_t)HSIC(\Theta_0, \Theta_0)}}$$

where

$$HSIC = \frac{\text{Tr}(\Theta_t L \Theta_0 L)}{(n-1)^2}$$

These kernel metrics can be readily computed from the trained models and extracted features.

**Capacity and effective geometry.** For more details on data-driven manifold capacity analysis, please refer to Appendix B.

**Setup of Figure 4a.** In Figure 4a, we showcase that the degree of feature learning is controlled by the effective learning rate $\bar{\eta}$ with the following standard setup:

- Data: Isotropic Gaussian manifolds with $R = 0.5, M = 15$.

- Model: We set $\sigma = \text{ReLU}, N = 1500, d = 1000, P = 100, K = 1$.

- Learning rule: We set $\ell = \ell_{MSE}, \eta = 50, c = 0$ and

$$\alpha = 10/128, 10/112, 10/96, 10/80, 10/64, 10/16, 10/4, 10/1$$

so that the normalized effective learning rates are

$$\bar{\eta} = 128, 112, 96, 80, 64, 16, 4, 1$$

which is computed by $\bar{\eta} = \frac{\eta \alpha^{-1}}{5}$ where the division by 5 normalizes the smallest $\eta \alpha^{-1}$ to be 1.

- Training: We trained the models for 100000 epochs with 40 repetitions per parameter combination.

- Plotting: We use sample mean and 95% confidence interval for each data point.

### D.3. Effective geometry reveals distinct learning dynamics

D.3.1. LEARNING STRATEGIES

**Compression strategy setup** In Figure 5b where the networks performs the compression strategy, we use a difficult-task setup with higher data manifold radius and more readout tasks:

- Data: Isotropic spherical manifolds with $R = 1.0, D = 8, M = 15$.

- Model: We set $\sigma = \text{ReLU}, N = 300, d = 200, P = 20, K = 27$.

- Learning rule: we set $\ell = \ell_{MSE}, \alpha = 1, c = 0$ and

$$\eta = 1, 5, 10, 20, 30, 40, 50, 60, 70, 80, 90, 100, 110, 120, 130, 140, 150$$

so that the normalized effective learning rates are

$$\bar{\eta} = 1, 5, 10, 20, 30, 40, 50, 60, 70, 80, 90, 100, 110, 120, 130, 140, 150.$$

- Training: We trained the models for 100000 epochs with 40 repetitions per parameter combination.

- Plotting: We use sample mean for each data point.

**Flattening strategy setup.** In Figure 5b where the networks performs the flattening strategy, we use an easy-task setup with smaller data manifold radius and very few readout tasks:

- Data: Isotropic spherical manifolds with $R = 0.5, D = 8, M = 15$.

- Model: We set $\sigma = \text{ReLU}, N = 300, d = 200, P = 20, K = 3$.

- Learning rule: we set $\ell = \ell_{MSE}, \alpha = 1, c = 0$ and

$$\eta = 80, 90, 100, 110, 120, 130, 140, 150, 160, 170$$

  so that the normalized effective learning rates are

$$\bar{\eta} = 80, 90, 100, 110, 120, 130, 140, 150, 160, 170.$$

- Training: We trained the models for 100000 epochs with 40 repetitions per parameter combination.

- Plotting: We use sample mean for each data point.

**Contour plot of learning strategies.** In Figure 5b and c, we use contour plots to visualize the different learning strategies adopted by the network. We use Equation 34 in (Chung et al., 2018) to approximate capacity using effective radius and dimension:

$$\alpha = \frac{1 + \left(\frac{1}{R_M^2}\right)}{D_M}$$

The scatter points with the same color correspond to a model trained with the same normalized effective learning rate $\bar{\eta}$ over different epochs.

### D.3.2. LEARNING STAGES

**Setup.** In Figure 5a, we adopt a setup with moderate radius and number of readout tasks that shows clean learning stages:

- Data: Isotropic spherical manifolds with $R = 1, D = 8, M = 15$.

- Model: We set $\sigma = \text{ReLU}, N = 300, d = 200, P = 20, K = 5$.

- Learning rule: we set $\ell = \ell_{MSE}, \eta = 10, \alpha = 1, c = 0$ so that the normalized effective learning rate is $\bar{\eta} = 10$.

- Training: We trained the models for 100000 epochs with 40 repetitions per parameter combination.

- Plotting: We use sample mean for each data point.

## E. Deep Neural Networks

### E.1. Experimental setup

In this section, we provide detailed information about the experimental setup for deep neural networks, including model architectures, datasets, training procedure, and manifold capacity measurements.

### E.1.1. MODELS

We use the VGG-11 models (Simonyan & Zisserman, 2015) for experimental results in the main paper. We also repeat these experiments on ResNet-18 (He et al., 2016). The specific implementation follows a similar setting in (Chizat et al., 2019) and is adapted from `https://github.com/edouardoyallon/lazy-training-CNN`.

**Output rescaling .**    As previously studied in (Chizat et al., 2019), multiplying the model outputs by a large scaling factor $\beta$ can induce lazy learning (we use the notation $\beta$ instead of $\alpha$ in (Chizat et al., 2019) to avoid confusion with the notation $\alpha$ as capacity in Equation (2) ). In this section, we use the inverse scaling factor $\beta^{-1}$ as the parameter to control the degree of feature learning. We define the *normalized effective learning rate* $\overline{\eta} = \beta^{-1}$. We also note several adjustments to the common training framework to adapt to using the inverse scaling factor $\beta^{-1}$ as the parameter to control the degree of feature learning.

- Rescaled loss function: To adjust for using the scaling factor $\beta$, we use the rescaled loss function $L_\beta = \frac{L}{\beta^2}$ with $L$ denotes the loss function to accommodate for the time parameterization of the loss dynamic for large $\beta$ as previously indicated in (Chizat et al., 2019) and (Geiger et al., 2020).

- Model's initial outputs as $0$: As mentioned in (Chizat et al., 2019), for the scaling factor $\beta$ to be able to control the rate of feature learning, the model output as initialization $f(W_0)$ must be equal $0$. To ensure this condition, we set $f(W_t) = h(W_t) - h(W_0)$ with $W_t$ be the model's weight at training step $t$, $h$ be the output of the network, and $f$ be the final adjusted network output.

**Number of repetitions.**    All model measurements (train accuracy, test accuracy, activation stability, etc.) are reported as the mean of 5 independently trained model (with different random seeds). The error bar indicates the bootstraped 95% confidence interval calculated using `seaborn.lineplot(errorbar=('ci', 95))`.

### E.1.2. DATASET

In this section, we list detailed information about the dataset used in the paper.

**CIFAR-10.**    The CIFAR-10 dataset (Krizhevsky & Hinton, 2009) consists of 60000 32x32 colour images in 10 classes, with 6000 images per class. There are 50000 training images and 10000 test images.

**CIFAR-100.**    The CIFAR-100 dataset (Krizhevsky & Hinton, 2009) is similar to CIFAR-10, except that it has 100 classes containing 600 images each. There are 500 training images and 100 testing images per class. Note that the images in CIFAR-10 and CIFAR-100 are mutually exclusive.

**CIFAR-10C.**    The CIFAR-10C dataset (Hendrycks & Dietterich, 2018) includes images from the CIFAR-10 evaluation set with common corruptions such as Gaussian noise, fog, motion blur, etc. The dataset has 15 different common corruption types, and 5 different severity levels for each corruption type.

### E.1.3. TRAINING PROCEDURE

- Loss function: We follow the theoretical results and practice used in (Chizat et al., 2019) to use mean-squared error loss to train all DNNs mentioned in the paper.

- Optimizer:    We    use    Stochastic    Gradient    Descent    with    momentum    (implemented    as `torch.optim.SGD(momentum=0.9)`) to train the models.

- Data augmentation: We apply the following data augmentation during training: `RandomCrop(32, padding=4)`, `RandomHorizontalFlip`.

- Learning rate and learning schedule: We follow the practice in (Chizat et al., 2019) and set initial learning rate $\eta_0 = 1.0$ for VGG-11 and $\eta_0 = 0.2$ for ResNet-18. The learning rate schedule is defined as $\eta_t = \frac{\eta_0}{1+\frac{1}{3}t}$.

- Initialization: We follow the practice in (Chizat et al., 2019) to initialize the model's weight using Xavier initialization (Glorot & Bengio, 2010) and the bias to be $0$.

- Batch size: We use batch size of 128 during training and batch size of 100 during evaluation.

E.1.4. MANIFOLD CAPACITY MEASUREMENTS

In this section, we provide detailed information about how we define object manifolds from the model's representations and measure the manifold capacity and geometric properties (Chung et al., 2018).

- Features extraction: For each image, we extract the object representation from the last linear layer (dimension 512) before the classification layer (dimension 10).

- Number of manifolds: We use 10 object manifolds for each measurement.

- Number of points per manifold: For each object manifold, we randomly sample 50 images from the interested class.

- Number of repetitions: Every capacity and geometry measurement is repeated 10 times per model instance (50 times if we have 5 model repetitions) and we report the mean and the error bar as the bootstraped 95% confidence interval calculated using `seaborn.lineplot(errorbar=('ci', 95))`.

### E.2. Capacity quantifies the degree of feature learning in deep neural networks

**Capacity and manifold geometry for VGG-11 models.** In Figure 4, we show manifold capacity along with other common metrics used to identify feature learning such as train accuracy, test accuracy, relative weight norm change, and activation stability. In this section, we provide other manifold geometric measurements along with manifold capacity in Figure 14.

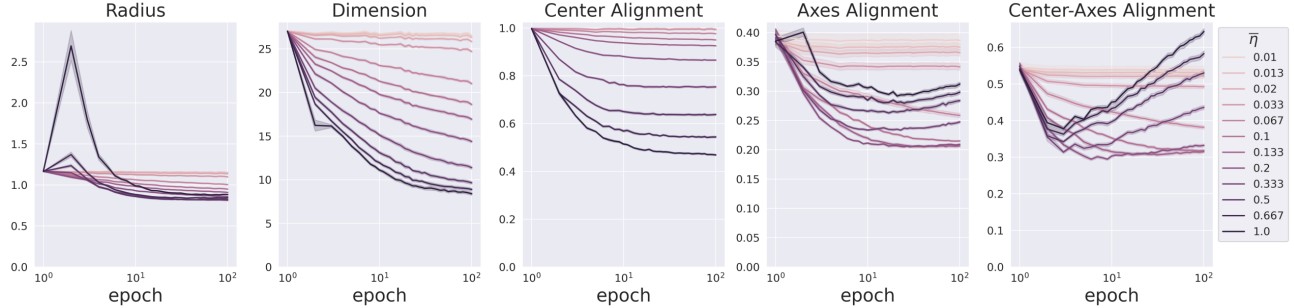

*Figure 14.* Manifold capacity and geometry for VGG-11 models trained with different $\overline{\eta}$

**Capacity quantifies the degree of feature learning in ResNet-18 models.** In section Section 3, we show that manifold capacity can capture the degree of feature learning in DNNs, specifically in VGG models. In this section, we empirically show this statement can also be extended to other model architectures, specifically ResNets, in Figure 15.

**Capacity quantifies the degree of feature learning in VGG-11 models trained with weight regularizer.** While most theoretical work in the lazy vs rich learning literature are formulated with vanilla mean squared error (MSE) loss (Jacot et al., 2018) (Chizat et al., 2019), in practice, MSE with weight regularizer (or weight decay) is used widely to prevent over-fitting and improve model generalization. In Figure 16, we explore the effect of weight decay to feature learning and demonstrate empirically that capacity can still quantify the degree of feature learning in models trained with L2-regularizer. We implemented L2-regularizer by setting `torch.optim.SGD(weight_decay=0.0002)`. We leave further study about the impact between the magnitude of weight regularizer and effective learning rate (and/or scaling factor) to the degree of feature learning as a potential future direction.

### E.3. Manifold capacity and manifold geometry delineate learning stages in deep neural networks

In section Section 4.2, we have demonstrated the use of effective manifold geometry to uncover hidden learning stages in 2-layer neural networks. In this section, we showed that using similar technique, we can also discover geometric learning stages in deep neural networks as well.

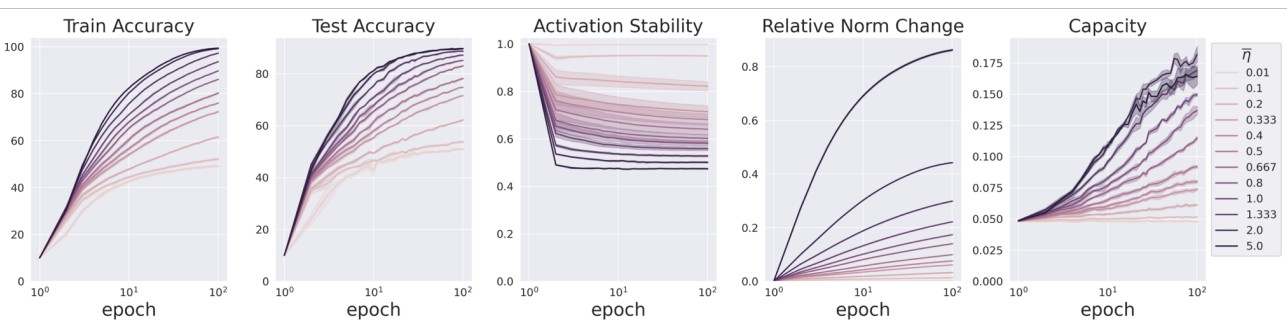

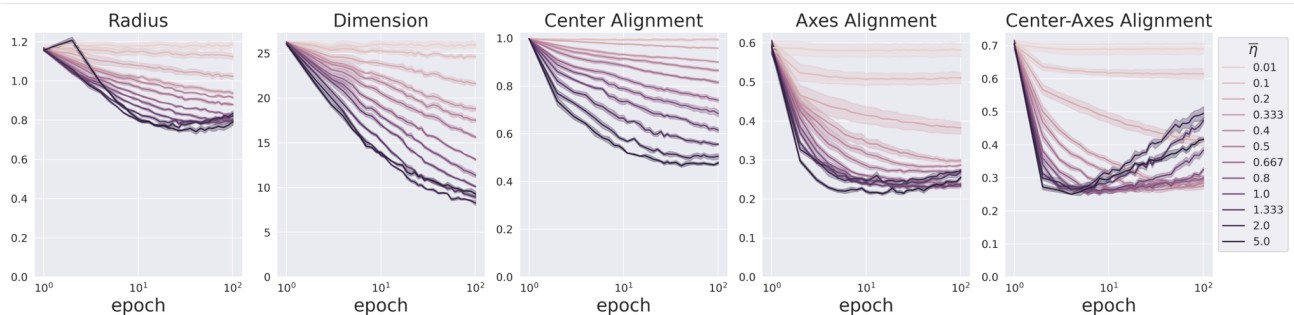

*Figure 15.* Manifold capacity and geometry of ResNet-18 models trained with different scale factor.

**Experiment setup** We used similar setup mentioned in Section E.1. In this section, to give a higher resolution into the learning dynamic, we extracted the model checkpoint at each training step (after each training batch, with `batch_size=100`) instead of each training epoch (after a whole train dataset iteration).

### E.4. Feature learning and downstream task: out-of-distribution generalization

In this section, we measure the performance of the models trained with different degree of feature learning (quantified by effective learning rate $\overline{\eta}$) on the downstream tasks for OOD using CIFAR-100, a dataset with no overlap with CIFAR-10, the dataset used to train the model.

#### E.4.1. EXPERIMENTAL SETUP

We use linear probe (Alain & Bengio, 2016) on representation from the last linear layer (dimension 512) to measure the performance of models trained on CIFAR-10 on the out-of-distribution dataset, CIFAR-100. Linear probes are linear classifiers trained on top of the representation to probe how much information the representations encode about a particular task or characteristic. This approach has been used widely in different fields including natural language processing (Belinkov et al., 2017) and computer vision (Raghu et al., 2021).

Here we provide detailed information about how we construct the linear probes.

**Optimizer.** We use *Adam* optimizer with initial learning rate $\eta_0 = 0.1$ and learning rate schedule is defined as $\eta_t = \frac{\eta_0}{1+\frac{1}{3}t}$. Other parameters are default `Pytorch` parameters.

**Number of epochs.** The linear probe is trained for 50 epochs, unless it is stopped early, as described by the early stop method below.

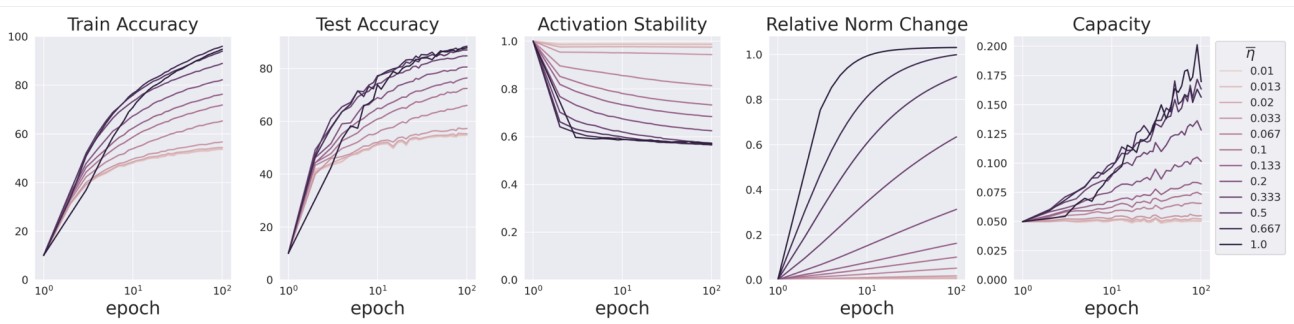

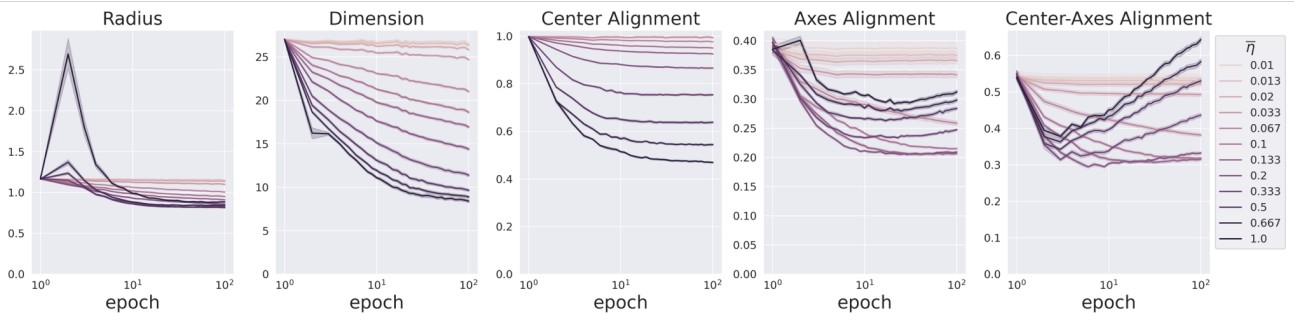

*Figure 16.* Manifold capacity and geometry of VGG-11 models with L2-regularizer trained with different scale factor.

**Early stop.** During training, if the validation loss is greater than the minimum validation loss so far for more than $N_{patience}$ epoch, then training is stopped. We set $N_{patience} = 3$.

### E.4.2. OOD PERFORMANCE FOR RESNET-18

In Section 5.2, we demonstrate how capacity and effective manifold geometry can be used to characterize the OOD performance of VGG-11 models trained with different effective learning rate $\bar{\eta}$. In this section, we show OOD performance and effective geometry of ResNet-18 models trained with different effective learning rate $\bar{\eta}$ in Figure 17. Interestingly, unlike VGG-11, for ResNet-18, the failure of models in the ultra-rich regime is characterized by the expansion of manifold dimension, not manifold radius.

## F. Recurrent Neural Networks

### F.1. Experimental Setup

In this section, we provide detailed information about the experimental setup for recurrent neural network in Section 5.1, including model architectures, datasets, training procedure, and manifold capacity measurements.

### F.1.1. DATASET

We used the package `neurogym` (Molano-Mazon et al., 2022) to simulate common cognitive tasks. In this paper, we trained recurrent neural networks to perform the following cognitive tasks: perceptual decision making, context decision making, and delay match sample. We followed the task configuration used in (Liu et al., 2024). We list detailed information of task configuration and descriptions below.

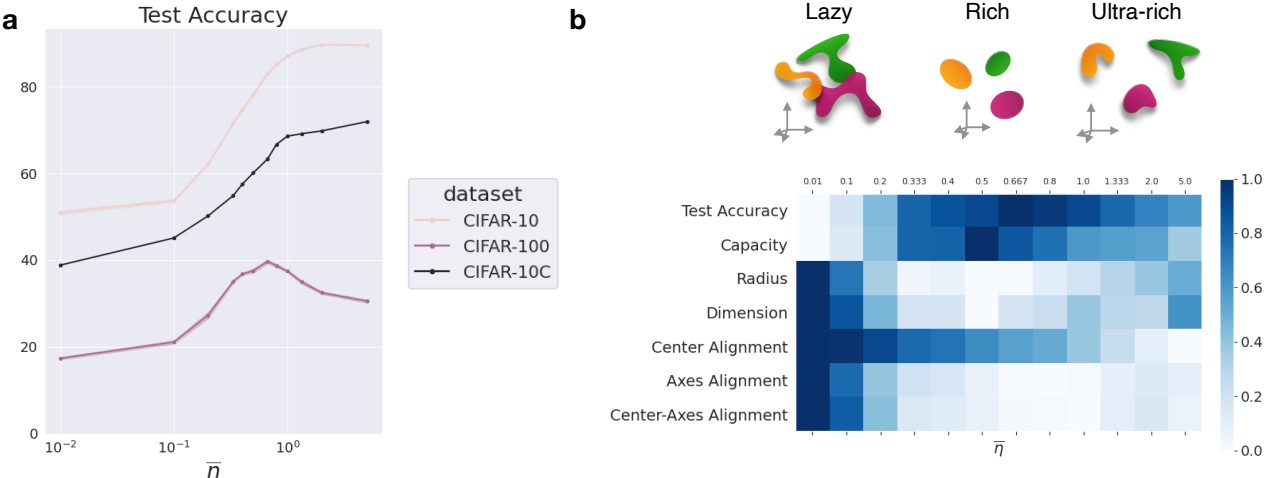

*Figure 17.* OOD performance and effective geometric measure of ResNet-18 models trained with different scale factor.

**Perceptual decision making** ([Britten et al., 1992](#)) ([documentation page](#))

- Task description: In each trial, given two noisy stimulus, the agent needs to integrate the stimulus over time to determine which stimuli has stronger signal.

- Task configuration: We set up the task using the following parameters: {timing: {fixation: 0, stimulus: 700, delay: 0, decision: 100}, dt: 100, seq_len: 8}

**Context decision making** ([Mante et al., 2013](#)) ([documentation page](#))

- Task description: In each trial, given two noisy stimulus, each has two modalities, the agent needs to integrate the stimulus in one specific modal while ignoring the other modal. The interested modal is given by the context.

- Task configuration: We set up the task using the following parameters: {timing: {fixation: 0, stimulus: 200, delay: 500, decision: 100}, dt: 100, seq_len: 8}

**Delay match sample** ([Miller et al., 1996](#)) ([documentation page](#))

- Task description: In each trial, a sample stimulus is shown during the sample period, which followed by a delay period. Afterwards, the test stimulus is shown. The agent needs to determine whether the sample and the test stimuli are matched.

- Task configuration: We set up the task using the following parameters: {timing: {fixation: 0, sample: 100, delay: 500, test: 100, decision: 100}, dt: 100, seq_len: 8}

F.1.2. MODELS

**Model architecture** We consider time-continuous recurrent neural networks (RNNs) architecture that are commonly used to model neural circuits ([Liu et al., 2024](#); [Ehrlich et al., 2021](#)). Specifically, we consider RNNs with 1 hidden layer, ReLU activation, $N_{in}$ input units, $N_{hidden}$ hidden units, and $N_{out}$ output unit. Let $x_t \in \mathbb{R}^{N_{in}}$, $y_t \in \mathbb{R}^{N_{out}}$ be the corresponding input and output at time-step $t$. The model's hidden representation $h_t$ and outputs $\hat{y}_t$ at time step $t$ can be defined by the given equations:

$$h_{t+1} = \rho h_t + (1 - \rho)(W_h \sigma(h_t) + W_i x_t) \tag{8}$$

$$\hat{y}_t = W_o \sigma(h_t) \tag{9}$$

In the above equation, $W_i \in \mathbb{R}^{N_{in} \times N_{hidden}}$, $W_h \in \mathbb{R}^{N_{hidden} \times N_{hidden}}$, $W_o \in \mathbb{R}^{N_{hidden} \times N_{out}}$. $\sigma(.)$ is the non-linear activation function, in which we used ReLU, and $\rho$ is the decay factor which is defined by $\rho = e^{\frac{-dt}{\tau}}$ with time step $dt$ and time constant $\tau$. We use $N_{hidden} = 300$ for all RNNs models.

**Weight rank initialization**  Following the practice in (Liu et al., 2024), we initialize the recurrence weight $W_h$ by initializing an initial full-ranked random Gaussian matrix, and then use Singular Value Decomposition to truncate the weight rank to the desired rank. The truncated weight matrix is then re-scaled to ensure that weight matrices with varying ranks have the same weight norm.

### F.1.3. TRAINING PROCEDURE

- Loss function: Since all three tasks that we consider are classification tasks, we use cross entropy loss.

- Optimizer: We use Stochastic Gradient Descent with momentum (implemented as `torch.optim.SGD(lr=0.003, momentum=0.9)`) to train the models.

- Batch size: We use batch size of 32 for each training step.

The models are trained for 10000 iterations and all models being compared achieved similar loss and accuracy after training (see Figure Figure 18, Figure 19, Figure 20 for more details).

### F.1.4. MANIFOLD CAPACITY MEASUREMENTS

In this section, we provide detailed information about how we define object manifolds from the model's representations and measure the manifold capacity and geometric properties (Chung et al., 2018).

- Features extraction: We extract the representation $h_t$ (in Equation Equation 8) from the hidden layer (dimension 300) with $t$ being the decision period of the trial.

- Number of manifolds: The number of possible choices in the decision period of all the three tasks that we consider is 2, so the number of manifolds are 2.

- Number of points per manifold: For each task-relevant manifold, we randomly sample 50 trials of the corresponding ground truth choices.

- Number of repetitions: Every capacity and geometry measurement is repeated 50 times and we report the mean and the error bar as the bootstraped 95% confidence interval calculated using `seaborn.lineplot(errorbar=('ci', 95))`.

### F.2. Additional results on other cognitive tasks

In section Section 5.1, we present the results on how the initial structural connectivity bias (initialized by varying the rank of the weight matrix) affects the feature learning regime and representational geometry of a given model in the perceptual decision making task (also called the two-alternative forced choice task) (Britten et al., 1992). In this section, we show more detailed results (including accuracy and loss) on the perceptual decision making task in Figure Figure 18, along with two other cognitive tasks, which are context decision making task (Mante et al., 2013) in Figure Figure 19 and delay match sample task (Miller et al., 1996) in Figure Figure 20.

## Perceptual Decision Making Task

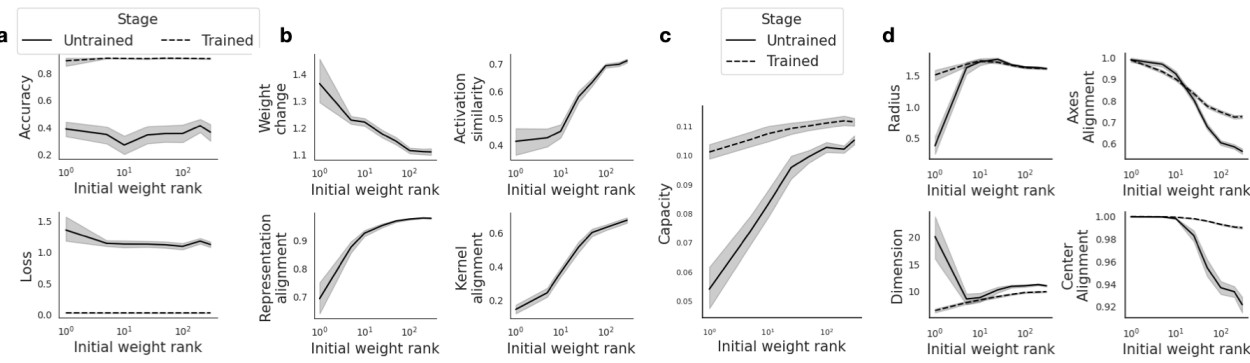

*Figure 18.* Structural connectivity bias in the two-alternative forced choice task. **a.** Model train and loss accuracy **b.** Weight change and alignment measurements **c.** Manifold capacity measurements **d.** Effective manifold geometry measurements.

## Context Decision Making Task

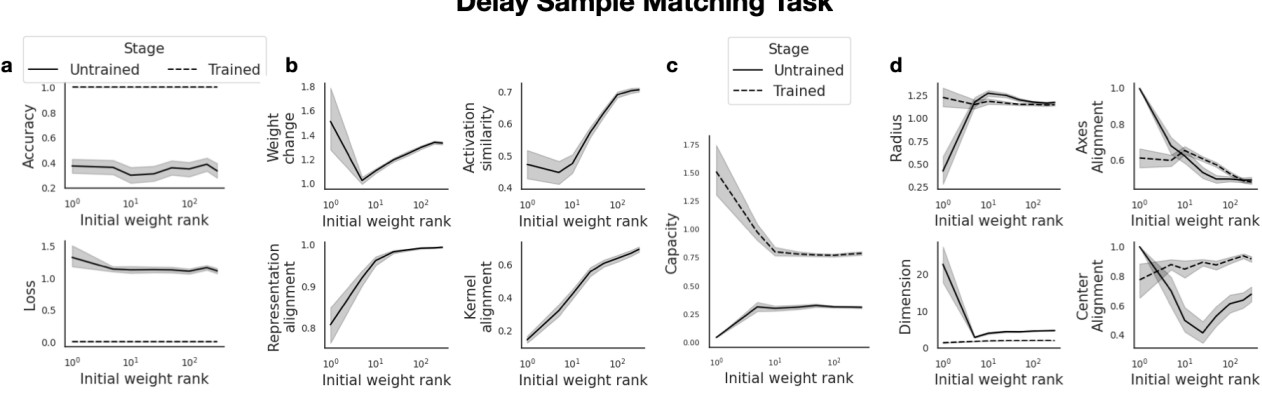

*Figure 19.* Structural connectivity bias in the context decision making task **a.** Model train and loss accuracy **b.** Weight change and alignment measurements **c.** Manifold capacity measurements **d.** Effective manifold geometry measurements.

## Delay Sample Matching Task

*Figure 20.* Structural connectivity bias in the delay mataching sample task. **a.** Model train and loss accuracy **b.** Weight change and alignment measurements **c.** Manifold capacity measurements **d.** Effective manifold geometry measurements.

