# OpenReview forum: "Feature Learning beyond the Lazy-Rich Dichotomy: Insights from Representational Geometry"
_ICML.cc/2025/Conference — ICML 2025 spotlightposter_

### Official Review · Reviewer_EQUL · 2025-03-10

**Overall Recommendation:** 3

**Summary:**

This paper introduces a framework to study subtypes of the rich regime during training of natural or biological neural networks.

Specifically, the authors use the known concept of manifold capacity to distinguish different phases of training and extract insights for ML and neuroscience.

## update after rebuttal

I thank the reviewers for their rebuttal, which partially answered my questions. The motivation of manifold capacity over manifold topology (and geonetry/dimension) is not convincing enough, as the authors state it is because it task-dependent, yet topology, geometry and dimension are also task-dependent. Hence, I maintain my score.

**Claims And Evidence:**

Until the method section, it seems that the analysis applies to any neural representation. However, it seems that it applies only to the last layer’s? Can the authors comment on the scope of the analysis? Similarly, it seems that the analysis only applies to classification tasks: can the authors comment on this too?

The authors mention distinguishing different stages of learning during training, but Figure 4c on that topic is hard to understand: it seems that capacity slowly and steadily increases during learning: how can the authors say that the four stages are “evident”? What would be an algorithm to automatically find these stages?

**Essential References Not Discussed:**

The  “conventional methods” to which the manifold capacity method is compared are not introduced in the related works in the main text.

See: “here we compare our method with several common measures for feature learning: accuracy curves, weight changes, and alignment methods (Table 1)” . *Why* choosing these methods for the comparison?

**Experimental Designs Or Analyses:**

In paragraph: Empirical justification in standard settings: I’m not sure that training two neural networks is enough to justify manifold capacity. Esp since both were trained in the lazy regime, which is not the point of this paper? Did I miss something?

Likewise, in Section 5, bold claims are made on the fact that geometric signatures can reflect DA and OOD behavior: but I’m not sure whether there are enough experiments to be able to make this point. Fig 6.c only show *one* instance where that is the case. How many neural networks were trained here? Do you have quantitative evidence that manifold capacity reflects DA and OOD across the models (and if so, could you comment on this in the main text?)?

**Methods And Evaluation Criteria:**

On Fig 3, if we zoom on the last epochs: could test accuracy differentiate between the regimes?

Section 4 only evaluates on two-layer neural networks. How about *evaluating* the framework on larger NNs, and comparing capacity to the other existing measures of NTK-label alignment, etc? (I understand that the framework is later *applied* to large NNs, but the values of the other measures -- such as NTK-label alignment -- are not given).

**Other Comments Or Suggestions:**

Is manifold dimension the extrinsic or intrinsic dimension?

Are the definitions of wealthy and poor regimes classical definitions? If so, could the authors add a reference?

Can you explain in more details what is the scale factor and why it represents the degree of feature learning? And unpack the link with the learning rate?

**Other Strengths And Weaknesses:**

Figures are excellent, but often hard to read.

Fig 2, eg, is gorgeous but *packed.*

Fig 4b is too small.

**Questions For Authors:**

NA

**Relation To Broader Scientific Literature:**

The choice of using manifold capacity could be further motivated in terms of what alternatives exist to study feature learning (and that could have potentially found similar learning stages?).

Eg dimensionality and frames: Kvinge et al. Internal Representations of Vision Models Through the Lens of Frames on Data Manifolds

Eg. Curvature: Acosta et al. Quantifying Extrinsic Curvature in Neural Manifolds.

Eg topology: Yoon et al. Tracking the topology of neural manifolds across populations

**Theoretical Claims:**

Theorem C.4 seems very important, since it could justify the use of capacity as a measure of feature learning. It should be in the main text. Generally, a lot of content is in appendix such that the main text is hard to read without referring to it constantly.

---

> ### Author Rebuttal · Authors · 2025-03-31
>
> We thank reviewer EQUL for their thorough evaluation and insightful questions. Below we address the reviewer questions on (1) our method, (2) experimental results. About the theoretical result, please refer to our response to reviewer NZt7, section 2, which address similar question.
>
> 1. Method and experiment setup
>
> a. `It seems that the method applies only to the last layer? Similarly, it seems that the analysis only applies to classification tasks`
>
> Our method is applicable to any hidden layer, we chose to focus on the last layer because it's typically where final learned features are examined. We’ll note in the paper that investigating intermediate layers is an interesting future direction. Our manifold capacity method focuses on classification tasks, building upon the classic perceptron storage capacity [Cover, 1965]. Extending the capacity notion to other tasks, such as regression, remains a promising area for future works.
>
> b. `The choice of using manifold capacity could be further motivated in terms of what alternatives exist to study feature learning (e.g, dimensionality and frames, curvature, topology)`
>
> We thank the reviewer for providing relevant geometric measures and will include these in our “Relevant works” section. Compared to other geometric approach, our method plays a unique role. Because these geometric measures are task-relevant, they allow us to more directly track the relationship between task-level performance and corresponding representational geometry (so the manifold dimension is extrinsic dimension, as it depends on the labels). In contrast, other geometric metrics, such as raw dimensionality, curvature, or topology, provide natural ways to characterize representations, but their connection to network performance remains more elusive.
>
> c. `Can you explain what is the scale factor and why it represents the degree of feature learning?`
>
> We use the standard setup in [Chizat, 2019] to use the scaling factor to tune the degree of feature learning (as also explained in reviewer NZt7 summary). For example, the original formula for MSE loss is $\frac{1}{2}(f(x)-y)^2$, with the output scaling factor $\alpha$, the loss would be $\frac{1}{2}(\alpha f(x) - y)^2$, which mean that we scale the network output by a scaling factor $\alpha$. The scaling factor can be used to tune the degree of feature learning as mentioned in [Chizat, 2019] section “Rescaled models” in p3 and Theorem 2.3 in p5. Intuitively, as the scaling factor grows, the network weights only need to change minimally while still achieving a big decrease in the objective loss, leading to lazy learning, in which the learned weight can be linearly approximated from the random initialized weight and not contain task-relevant features.
>
> d. `The “conventional methods” to which the manifold capacity method is compared are not introduced in the related works. Why choosing these methods?`
>
> We thank the reviewer for pointing this out! Weight changes, test accuracy are two measures used in the first lazy-rich paper [Chizat 2019]. NTK-label alignment and Representation-label alignment are measures used in follow-up works [Geiger 2020] and [Kumar 2024]. We will put these citations in our updated version.
>
> 2. Experimental results:
>
> a. `Fig 4c: distinguishing different stages of learning during training`
>
> Fig. 4c shows that while the performance (e.g capacity) monotonically increases (1 learning stage), the geometric measures can capture more subtle change in the representations during training, and reflect distinct learning stages (as acknowledge by reviewers guAP, sxs9, NZt7). About the concern on robustness of this finding, please refer to our response to reviewer NZt7, section 1.
>
> b. `Empirical justification in standard settings: I’m not sure that training two neural networks is enough to justify manifold capacity`
>
> We showed empirically that capacity can correctly track the ground-truth of feature learning (measured by the inverse scaling factor $\bar\eta$) across different settings, including simple 2-layer models (Fig. 3) and deep nets (VGG-11: Fig. 2a,b, Fig. 13, ResNet18: Fig 14).
>
> We want to clarify that each reported data point resulted from 5 models initialized with different random seeds. We also vary the feature learning rate $\bar\eta$ from 0.01 to 1 with 10 different values to train our models, therefore including both rich and lazy regimes. For each model architecture, our results are reported from training 50 neural networks.
>
> c. `In Section 5, bold claims are made that geometric signatures can reflect DA and OOD behavior. Fig 6.c only show *one* instance where that is the case. How many neural networks were trained here?`
>
> For the DA and OOD experiments, we used two distinct architectures VGG-11 (Fig 6) and ResNet18 (Fig 16), and indeed we've observed capacity can capture OOD performance and distinct geometric signatures with different architectures. About the number of trained neural nets , we refer to the answer on 2b.

---

### Official Review · Reviewer_NZt7 · 2025-03-12

**Overall Recommendation:** 4

**Summary:**

The authors revisit the dichotomy of *lazy learning*, where neural networks do not learn data-dependent features and instead act essentially as a kernel machine, and *feature learning*, where they do. They argue that there are in fact several distinct learning regimes in the rich regime, which they untangle by studying the the geometric properties of task-relevant manifolds, for example the point cloud of neural activations corresponding to stimuli in a given class for a classification task.

They study experimentally two-layer and deep neural networks, tuning the amount of feature learning by varying the inverse scale factor $\eta$ à la Chizat 2019. They show that various geometric measures of manifold geometry are connected to the prediction error (but unfortunately do not even state the result in the main text). They then go on to show experimentally that various measures of manifold geometry correlate well with feature learning on synthetic data (Sec 3) and that different measures of manifold geometry vary at different stages in training, which the authors call the clustering, structuring stage, separating stage and stabilizing stage.

Finally, they apply their methodology to recurrent neural networks trained on tasks studied in theoretical neuroscience. They confirm previous work showing that the rank of the initial connectivity governs whether the learning dynamics are lazy or rich, but find that RNNs achieve roughly the same manifold capacity either way.

## After the discussion...
... I increased my score, as I explained in my rebuttal comment.

**Claims And Evidence:**

See Strengths & Weaknesses.

**Essential References Not Discussed:**

See above.

**Experimental Designs Or Analyses:**

See Strengths & Weaknesses.

**Methods And Evaluation Criteria:**

See Strengths & Weaknesses.

**Other Comments Or Suggestions:**

- This is a small point, but I would not describe the (great!) paper of Chizat, Oyallon & Bach '19 as demonstrating "that neural networks can perform well even when there are negligible changes in the weights of the networks" (p. 1) Instead, they showed that two-layer neural networks whose first-layer parameters move only a little are instead severely limited on high-dimensional tasks compared to feature learners in the same way that kernels are, therefore coining the term "lazy".
- When citing books, please give a more specific reference to a subsection or a theorem, otherwise the reference is useless (for example when citing the book by Vershynin.

**Other Strengths And Weaknesses:**

The article presents an interesting and really well-written exploration of how various quantities describing the geometry of the task manifold evolve during the training dynamics of neural networks. These geometric quantities show promise in that they are related to the accuracy via a theoretical result; however, since this result is not even stated in the main text, and the discussion in the supplementary material spans several additional pages, I do not want to consider it too heavily in this review out of respect for the page limit.

Another strength of the paper is that it is packed with different concepts to analyse neural networks - but that is also a weakness. It is hard to track all the different quantities relating to manifold geometry; for me for example it is not clear how independent these measures are, and to what extent one can explain the other. Finally, it is not clear at all how they relate to accuracy; I guess the theoretical result should fill in that void, but then it should be discussed in the main text.

An interesting observation that the authors make is that they identify four different stages of the rich regime (line 340ff) However, this appears to be a one-time observation in an experiment, and unfortunately, the authors do not explore it further, try to reproduce it in another setting, or further analyse what to me looks like the main result of this study (and it is indeed one of the three main results mentioned by the authors in their introduction) I therefore find the overall amount of results a bit lacking to recommend acceptance, but I think it is worthwhile to explore this observation further. I would make acceptance contingent on further expansion of this observation. While it is obviously not for myself to decide, I wonder whether this paper in a longer form published in a good journal like JMLR wouldn't do more justice to its content.

**Questions For Authors:**

No further questions.

**Relation To Broader Scientific Literature:**

Broadly speaking, the paper does a good job of relating to the broader scientific literature. A couple of points should be improved:

- The approach of the paper is really a (well-executed!) application of the manifold untangling hypothesis that originates in neuroscience. The authors mention DiCarlo & Cox (2007) in a footnote, but I think this should be stated more prominently in the main text, for example at the beginning of section 1.1.
- There were clearer and earlier demonstration of the advantage of feature-learners over lazy models, including Ghorbani et al. NeurIPS 2019 and 2020; Daniely & Malach NeurIPS 2020; and Refinetti et al. ICML 2021 (start of p. 2)

**Theoretical Claims:**

The authors report a theoretical result which sounds intriguing - connecting various geometric quantities to prediction error. However, the authors do not even state this result in the main text, for reasons that are not clear to me. If it is important, it should be stated there!

---

> ### Author Rebuttal · Authors · 2025-03-31
>
> We thank reviewer NZt7 for their thoughtful reviews and suggestions. We appreciate the reviewer recognized that our works “presents an interesting and really well-written exploration of how various quantities describing the geometry of the task manifold evolve during the training dynamics of neural networks”. We greatly value the reviewer’s insightful questions about (1) the robustness of our learning stage results, (2) the presentation of the theoretical results, (3) the relationship between our different geometric measurements, as we agree that addressing these concerns would greatly strengthen the paper. Below we address these well-thought questions and concerns:
>
> 1.  `An interesting observation that the authors make is that they identify four different stages of the rich regime (line 340ff) However, this appears to be a one-time observation in an experiment, and unfortunately, the authors do not explore it further, try to reproduce it in another setting`
>
>     We appreciate that reviewer recognize that our geometric measures can offer a deeper understanding on learning dynamic. About the concern about the robustness of the “learning stages” findings in Fig. 4c, we want to clarify that:
>
>     1. **Number of random seeds repetitions:** All our empirical experiments contain results from 5 different models repetitions, initialized from different random seeds, and the heat map in Fig. 4c is the results averaged from 5 repetitions. While we did include the information about number of repetitions in Appendix section E.1, we admit that this is an important information and should have been mentioned in the caption figure, and we will include this information in the final version of the paper.
>     2. **Different models configurations:** While Fig. 4c focuses on a single model configuration (VGG-11 with $\bar\eta=0.2$) to demonstrate a specific example of how our methods can offer deeper insights on learning stages, we indeed have experimental results that similar hidden stages can be observed across various richness degrees (across $\bar\eta$ values) and model architectures (VGG-11 and ResNet-18) at https://imgur.com/a/learning-stages-figures-f5pZEWa. Since reviewer NZt7 mentioned that it is worthwhile to explore this observation further, we sincerely hope that the reviewer can take a look at these experiment results, which shows that our geometric measurements can reveal hidden stages in various settings of richness degrees and model architectures.
> 2. `The authors report a theoretical result which sounds intriguing - connecting various geometric quantities to prediction error. However, the authors do not even state this result in the main text`
>
>     We thank the reviewer for the careful reading of our theoretical results. We fully agree that an analytical characterization of capacity throughout training offers valuable insights that help justify our approach. Due to page limitations, we made the difficult decision to place the detailed theoretical analysis in the appendix, as we wanted to emphasize the breadth of our empirical findings in the main text. In the final version of the paper, we will include more details on the theoretical results in the main text to better highlight their significance.
>
> 3. `It is hard to track all the different quantities relating to manifold geometry; for me for example it is not clear how independent these measures are, and to what extent one can explain the other.`
>
>     Previous work on manifold capacity and its effective geometric measures [Chou et al., 2024] offers comprehensive explanations, covering theoretical foundations, numerical analyses, and intuitive insights, on how to interpret these measures and their relationship to capacity in the supplementary material. We have incorporated some relevant parts into our own appendix. In the final version of the paper, we will include additional examples to better illustrate the independence among these measures. In particular, we will provide a mathematical explanation on how effective dimension interact with axis alignment (higher axis alignment would effectively reduce the manifold dimension, because the variation subspaces become more overlapped); how effective radius interact with center alignment (higher center alignment would effectively increase the radius, because the manifolds become closer to each other).
>
>     References:
>
>     1. Chou, Chi-Ning, et al. "Neural manifold capacity captures representation geometry, correlations, and task-efficiency across species and behaviors." *bioRxiv* (2024).
>
> We thank reviewer **NZt7** once again for their time, insightful questions, and actionable suggestions that help us to strengthen both the presentation of our theoretical results and the robustness of our experimental results! We hope our responses address your concerns. Please let us know if there’s any other details that we can further clarify. Thank you very much for your time and consideration!

---

> > ### Comment · Reviewer_NZt7 · 2025-04-03
> >
> > I have read the comments of the authors. I appreciate the effort to run additional experiments, and I strongly recommend the authors move the statement of the theoretical result to the main text with the additional space that is afforded to them. I have therefore increased my rating.

---

### Official Review · Reviewer_sxs9 · 2025-03-14

**Overall Recommendation:** 4

**Summary:**

Numerous studies in representation learning have been conducted to evaluate the quality of features learned by DNNs, particularly in determining whether a neural network functions within the lazy or rich regime. In this paper, the authors presented theoretical foundations grounded in manifold capacity theory to address the Lazy vs. Rich dichotomy issues, examining feature learning through the geometric properties of task-relevant manifolds. Additionally, the paper highlighted that the training of neural networks evolves through distinct learning stages, as reflected by the dynamics of manifold geometry. It also identified emerging learning strategies as networks demonstrate varying levels of richness in their learning. With robust theoretical underpinnings and empirical support, the proposed geometric features provide a valuable tool for evaluating the depth of feature learning.

## Update after rebuttal
Thank you to the authors for their clarifications. I look forward to reviewing the updated manuscript and will maintain my current score.

**Claims And Evidence:**

- C1. The paper is well-motivated, and the experimental setups clearly demonstrate the effectiveness of the proposed method. Drawing on the theoretical definitions of feature manifolds, the authors provide empirical evidence, including variations in manifold capacities in relation to the Lazy vs. Rich regime, compared to traditional measures (Figure 2). Additionally, the comparison of input dimensions and their impact on manifold capacities is particularly intriguing (Figure 3).
- C2. The use of manifold geometry to illustrate learning strategies and stages is effectively demonstrated (Figs. 4 and 6). This approach offers substantial potential as an interpretable tool for elucidating the model's learning dynamics, with the added benefit of enhancing the model's generalizability and robustness.

**Essential References Not Discussed:**

In terms of comparison with other conventional measures, I believe the authors have sufficiently addressed the relevant references, and no critical omissions require attention.

**Experimental Designs Or Analyses:**

- E1. Overall, the authors' experimental designs and analyses were well-structured and rigorously evaluated. For instance, the experiments measuring the degree of feature learning (Figs. 2 and 3) effectively demonstrate a strong alignment between manifold capacity and the Lazy vs. Rich regimes, particularly when compared to other conventional metrics, which is persuasive. Furthermore, Figure 4 presents consistent results using manifold radius and dimension across different regimes.

**Methods And Evaluation Criteria:**

- M1. All definitions are grounded in manifold capacity theory, including the computation of dimension, radius, and various alignments, and are thoroughly evaluated within Lazy vs. Rich regime frameworks on both synthetic and image datasets. The experiments involving RNNs, as well as those focused on domain adaptation and out-of-distribution generalization, are intriguing and offer great insights.

**Other Comments Or Suggestions:**

I have no other comments or suggestions.

**Other Strengths And Weaknesses:**

- S1. Overall, the paper is well-structured and presents the experimental results in an intuitive and accessible manner. The figures illustrating the correlations between geometric characteristics and capacity are particularly valuable in enhancing understanding.

**Questions For Authors:**

I have listed my questions in each section.

**Relation To Broader Scientific Literature:**

Feature learning is a fundamental aspect of neural network research in machine learning, extending far beyond the simplistic lazy-versus-rich dichotomy. Understanding the relationship between feature learning and performance is essential for designing network architectures and learning algorithms that offer high reliability and transparency for practical applications. The authors' proposed method, grounded in manifold capacity, holds significant potential as an interpretable tool that aligns well with the current research direction.

**Theoretical Claims:**

- T1. One of the questions concerns the definitions of simulated manifold capacity and packability presented in the main text. According to Equation 1 and Section 2.1, the authors employed a random projection \Phi from R^N to R^n. Further details about this process are needed. Specifically, is this projection intended for dimensionality reduction? What type of random weights were utilized? Additionally, does the use of random weights always guarantee the identification of the same feature manifold? (This question addresses the identifiability of manifolds.)

---

> ### Author Rebuttal · Authors · 2025-03-31
>
> We thank reviewer **sxs9** for their thorough evaluation of our paper’s motivation, methods, and results, as well as the valuable feedback and insightful questions! We greatly appreciate the reviewer for recognizing that our work “offers substantial potential as an interpretable tool for elucidating the model's learning dynamics” and “the experimental setups clearly demonstrate the effectiveness of the proposed method”. Below we address the reviewer questions about simulated manifold capacity and missing details in the Appendix.
>
> 1. `According to Equation 1 and Section 2.1, the authors employed a random projection \Phi from R^N to R^n. Further details about this process are needed.`
>
>     We thank the reviewer for pointing out the missing details about random projection in Equation 1, which specifies the definition of simulated manifold capacity. Intuitively, simulated manifold capacity measures the smallest dimensional subspace such that the projected manifolds is linearly separable with probability ≥ 0.5, under the distribution of random binary label dichotomy and random projection. The random projection matrix can be sampled from the standard normal distribution (then normalized to ensure unit norm). Therefore, simulated capacity can be computed numerically by performing a bisection search to find the smallest dimensional subspace such that the probability of manifold separability ≥ 0.5. Further details can be found in [Cohen et al. 2020], p11, section “Measuring capacity numerically from samples”). Below we provide the pseudocode to compute simulated capacity, which will also be included in the final version of the paper
>
>     **Pseudocode:**
>
>     Step 1: Set `min_dim` (minimum subspace dimension for random projection, usually 2) `max_dim` (maximum subspace dimension, usually the original dimension),  `num_repetition` (number of random repetitions to sample the binary dichotomy and random projection matrix), `tolerance` (error tolerance between the target, which is the 0.5 probability value, and the found value), and `max_iteration` .
>
>     Step 2: Set `mid_point = min_dim + np.floor((max_dim-min_dim)/2)` . Estimate the probability of linearly separable for `mid_point` , or `f(mid_point)`
>
>     Step 2.1: To estimate the linearly separable probability for `mid_point` , we sample random projection matrix `M` and binary label `y` for `num_repetition` times. For each repetition, we can use quadratic optimization to determine whether the current projected manifold sample is linearly separable (0 or 1). The returned estimated probability is the ratio between the number of linearly separable samples over the total number of repetitions.
>
>     Step 3: While `abs(f(mid_point) - 0.5) > tolerance` and `current_iteration < max_iteration` , update `mid_point`, and repeat the computation of `f(mid_point)` until reach the value within tolerance, or the maximum number of iterations.
>
>     Step 3.1: If `f(mid_point) > 0.5` , update `max_dim = mid_point`, else update `min_dim = mid_point` . Store tuple `(mid_point, f(mid_point))`
>
>     Step 4: Return `mid_point` if within `tolerance`, else use interpolation to estimate the value of `mid_point` such that `f(mid_point) = 0.5`
>
>     References:
>
>     1. Cohen, Uri, et al. "Separability and geometry of object manifolds in deep neural networks." *Nature communications* 11.1 (2020).
> 2. Supplementary Material: `There were a few minor missing details, such as the type of initialization for random Fourier features and the specifics of the teacher-student setting in Section C.`
>
>     We thank the reviewer for pointing out the minor missing details. We follow the same setting as in [Montanari et al. 2019] and [Ba et al., 2022]. Specifically, the initial weights of the 2-layer network were sampled independently from isotropic Gaussians (i.e., the random features are orthogonal with each other with high probability), this is also described in item 2 of Assumption C.1 in the appendix. As for the teacher-student setting, the teacher is modeled as a hidden direction $\beta^*$ and examples data $x_1,\dots,x_{n_{\text{train}}}$ are generated indenpedently from isotropic Gaussians with labels being $y_1,\dots,y_{n_{\text{train}}}$ where $y_i=1$ with probability $F(\langle\beta^*,x_i\rangle)$ for some monotone function $F(\cdot)$.  This is also discussed in Setting C.2 in the appendix and we will provide more details in the updated version of the paper.
>
>     References:
>
>     1. Montanari, Andrea, et al. "The generalization error of max-margin linear classifiers: High-dimensional asymptotics in the overparametrized regime." *arXiv preprint (2019).
>     2. Ba, Jimmy, et al. "High-dimensional asymptotics of feature learning: How one gradient step improves the representation." *Advances in Neural Information Processing Systems* 35 (2022).
>
> We thank reviewer **sxs9** once again for their time, insightful questions, and actionable feedback!

---

### Official Review · Reviewer_guAP · 2025-03-14

**Overall Recommendation:** 4

**Summary:**

This paper uses manifold capacity measures to assess neural representations and learning in the rich and lazy learning regimes. The authors show, both numerically and, in some cases, analytically, that these manifold capacity measures provide a deeper understanding of learning dynamics and neural representations that are beyond traditional metrics like accuracy, weight changes, label alignment, and representational alignment. Novel insights include, that representations differ when the network starts with either a beneficial (wealthy) or poor (decremental) initial weight structure; and that learning unfolds in four distinct phases: clustering, structuring, separating, and stabilizing, which vary depending on the learning regime. The authors further extend their analysis to recurrent neural networks, exploring how hidden-layer representations evolve under high- and low-rank initializations and further investigate out-of-distribution generalization, showing how the the rich and lazy regimes influences the performance depending on the underlying out-of-distribution-tasks' complexity.

**Claims And Evidence:**

The authors claim that manifold capacity measures provide an improved and more detailed understanding of the rich and lazy learning regimes. All specific claims (outlined in the summary) are supported by strong (analytical and) numerical evidence. The analysis is both thorough and comprehensive. The figures are clear and effectively visualize the evidence, providing strong support for the claims.

**Essential References Not Discussed:**

Woodworth, Blake, et al. "Kernel and rich regimes in overparameterized models." Conference on Learning Theory, PMLR, 2020. This paper is one of the first to systematically study the relationship between the rich and lazy regimes and generalization.

**Experimental Designs Or Analyses:**

See Methods And Evaluation Criteria

**Methods And Evaluation Criteria:**

The methods and evaluation criteria, including the choice of network architectures and datasets, are well justified. They cover a range of cases, from the simpler and more tractable 2-layer nonlinear ANNs and point clouds to more realistic architectures like ResNet and datasets such as CIFAR. Additionally, the tasks and scope of the recurrent neural network studies are well justified and align with established standards.

**Other Comments Or Suggestions:**

The text, labels, and ticks in Figures 2 and 4 are often too small and should be made more legible.

There is a duplication of text: "We adopt the setting from previous work (Liu et al., 2024) on investigating how differences in connectivity initialization affect the learning process." and "To study how connectivity structure impacts learning strategies, we follow the setup in (Liu et al., 2024)...". This repetition could be avoided by consolidating the statements.

I think the following paragraph:  "In a network that does not learn task-relevant features (e.g., lazy learning, random features, Figure 1b, left), the manifolds are poorly organized, making them harder to distinguish (e.g., smaller margin, smaller solution volume). In contrast, when a network learns task-relevant features (e.g., rich learning Figure 1b, right), the manifolds become well-organized and easier to separate (e.g., larger margin, larger solution volume)." may be a little bit counterintuitive as the lazy and rich learning regimes are generally associated with fast exponential and slow step-like learning dynamics. Maybe the authors can elaborate on that, e.g. it is easy to find a linear searation in a high-dimensional random projection of the data which leads to exponentially fast learning, however also leads to poorly organised (or not even really to any) manifolds.

**Other Strengths And Weaknesses:**

The paper is very dense but well written.

**Questions For Authors:**

N/A

**Relation To Broader Scientific Literature:**

The presented work uses manifold capacity measures to study representation learning in artificial neural networks. In theory, this approach could also be applied to examine interindividual differences in neural representations in neuroscience, suggesting that individuals may differ in their operation within the rich and lazy regimes. Furthermore, the analysis pipelines and measures presented could enhance our understanding and comparison of representation learning both in neural networks and in the brain during learning.

**Theoretical Claims:**

I did not verify the theoretical claim in Appendix C.

---

> ### Author Rebuttal · Authors · 2025-03-31
>
> We thank reviewer **guAP** for spending time and effort to thoroughly read, evaluate, and provide detailed comments and suggestions for our manuscript!
>
> We greatly appreciate the reviewer for recognizing that our works provide `a deeper understanding of learning dynamics and neural representations that are beyond traditional metrics like accuracy, weight changes, label alignment, and representational alignment` and the claim is supported by `strong (analytical and) numerical evidence`!
>
> We also thank the reviewers for excellent advices on (1) adding the missing references to Woodworth, Blake, et al., (2) improving the legibility of text, labels, and ticks in Figures 2 and 4, and (3) providing more context on how to relate “exponentially fast learning” in the lazy regime with the “poorly organized manifold” concept. We really values the reviewer’s suggestions and have added action items to incorporate these points to our updated version!
>
> We thank reviewer **guAP** once again for their time and actionable feedback!

---

### Decision · Program_Chairs · 2025-05-01

**Decision:**

Accept (spotlight poster)

**Comment:**

This paper presents a compelling and novel framework for understanding feature learning dynamics in neural networks via the geometry of task-relevant manifolds. It moves beyond the standard lazy vs. rich dichotomy by characterizing learning stages through manifold capacity and related geometric descriptors.

The reviewers found the work both theoretically insightful and empirically well-supported, noting its relevance across domains, including neuroscience and machine learning. In particular, the identification of learning stages (clustering, structuring, separating, stabilizing) via representational geometry is a notable contribution. The experiments are diverse and carefully designed, covering synthetic and image datasets, and RNNs, and include settings involving OOD generalization.

While several reviewers raised valid concerns, such as the placement of theoretical results in the appendix, the robustness of learning stages, and the motivation for choosing manifold capacity over alternative measures, these were well addressed in the rebuttal.

Given the paper’s originality, technical depth, and relevance to both theoretical and applied ML communities, I support acceptance to ICML.